# VCF1 is a p97/VCP cofactor promoting recognition of ubiquitylated p97-UFD1-NPL4 substrates

Ann Schirin Mirsanaye[1,4], Saskia Hoffmann[1,4], Melanie Weisser [1], Andreas Mund [1], Blanca Lopez Mendez [1], Dimitris Typas[1], Johannes van den Boom [2], Bente Benedict[1], Ivo A. Hendriks [1], Michael Lund Nielsen [1], Hemmo Meyer [2], Julien P. Duxin [1], Guillermo Montoya [1] & Niels Mailand [1,3] ✉

The hexameric AAA+ ATPase p97/VCP functions as an essential mediator of ubiquitin-dependent cellular processes, extracting ubiquitylated proteins from macromolecular complexes or membranes by catalyzing their unfolding. p97 is directed to ubiquitylated client proteins via multiple cofactors, most of which interact with the p97 N-domain. Here, we discover that FAM104A, a protein of unknown function also named VCF1 (VCP/p97 nuclear Cofactor Family member 1), acts as a p97 cofactor in human cells. Detailed structure-function studies reveal that VCF1 directly binds p97 via a conserved α-helical motif that recognizes the p97 N-domain with unusually high affinity, exceeding that of other cofactors. We show that VCF1 engages in joint p97 complex formation with the heterodimeric primary p97 cofactor UFD1-NPL4 and promotes p97-UFD1-NPL4-dependent proteasomal degradation of ubiquitylated substrates in cells. Mechanistically, VCF1 indirectly stimulates UFD1-NPL4 interactions with ubiquitin conjugates via its binding to p97 but has no intrinsic affinity for ubiquitin. Collectively, our findings establish VCF1 as an unconventional p97 cofactor that promotes p97-dependent protein turnover by facilitating p97-UFD1-NPL4 recruitment to ubiquitylated targets.

The highly abundant and evolutionarily conserved AAA+ ATPase Valosin-containing protein (VCP)/p97 (known as Cdc48 in lower eukaryotes) is a crucial mediator of ubiquitin-dependent processes and proteostasis mechanisms in eukaryotic cells, impacting and regulating most aspects of cell physiology[1,2]. The action of p97 principally entails the threading of, in most cases, ubiquitin-modified client proteins through the central pore of the barrel-shaped p97 homohexamer via two hexameric ATPase rings formed by the D1 and D2 ATPase domains within each p97 monomer. This leads to the unfolding of the substrate protein, enabling its degradation by the 26S proteasome or extraction from macromolecular complexes or cellular membranes[3–7]. The critical importance of p97 in cell and organismal physiology is underscored by the association of p97 mutations with neurodegenerative disorders including inclusion body myopathy with Paget's disease of bone and frontotemporal dementia (IBMPFD) and amyotrophic lateral sclerosis (ALS)[8]. On the other hand, targeting increased levels of proteotoxic stress in cancer cells using small molecule inhibitors of p97 represents a promising therapeutic

[1]Novo Nordisk Foundation Center for Protein Research, University of Copenhagen, DK-2200 Copenhagen, Denmark. [2]Molecular Biology I, Faculty of Biology, University of Duisburg-Essen, 45117 Essen, Germany. [3]Center for Chromosome Stability, Department of Cellular and Molecular Medicine, University of Copenhagen, DK-2200 Copenhagen, Denmark. [4]These authors contributed equally: Ann Schirin Mirsanaye, Saskia Hoffmann. ✉e-mail: niels.mailand@cpr.ku.dk

avenue[9]. A detailed understanding of the workings of the p97 system is therefore of considerable biomedical importance.

Around 40 established cofactors are responsible for the interplay between p97 and its numerous cellular substrates, governing substrate recognition and specificity, subcellular localization and/or enzymatic activity of p97 complexes[2,10,11]. These cofactors directly interact with p97 via one or more well-defined domains and motifs, most of which recognize the p97 N-domain. Known p97 N-domain-binding modules include both globular UBX and UBX-like domains, which adopt a ubiquitin-like fold and interact with a large surface area on the N-domain, and short linear motifs including the SHP box and the VIM and VBM motifs[11]. While the SHP box binds the p97 N-domain close to the central pore and the D1 ATPase domain, the VIM and VBM motifs adopt α-helical folds that enable them to dock into the hydrophobic cleft formed between the N- and C-terminal lobes of the p97 N-domain[11,12]. A considerable proportion of disease-associated p97 mutations map to the N-domain, highlighting the importance of p97-cofactor interactions for the proper functioning of the p97 system[8]. A smaller number of cofactors contain PUB or PUL domains that interact with the p97 C-terminus[11,12]. Many p97 cofactors also harbor one or more ubiquitin-binding domains that target p97 to ubiquitylated substrates, most of which are modified by K48-linked polyubiquitin chains[13]. In addition, some p97 cofactors contain enzymatic domains that impart further relevant functionalities such as ubiquitin chain editing activity to p97 complexes[12].

p97-cofactor interactions generally appear to be transient and highly dynamic[14], and while some major cofactors bind p97 in a mutually exclusive manner, different cofactors in many cases assemble jointly on individual p97 hexamers[12]. The regulatory mechanisms and hierarchies underlying the formation and dissociation of complexes between p97 hexamers and combinations of cofactors are not well understood, hampered by the complexity of the p97 system and the reliance on in vitro studies with purified proteins for characterizing this interplay. However, a general theme is that the basic functionality of p97-dependent processes is driven by primary p97 cofactors and augmented by accessory cofactors[15]. The heterodimeric UFD1-NPL4 complex defines one of the most important and best characterized primary p97 cofactors with a key role in unfolding ubiquitylated substrates. Recent elegant studies have provided detailed structural insight into how UFD1-NPL4 recruits p97 to ubiquitylated targets via multiple ubiquitin-binding domains, collaborating with p97 in a highly integrated fashion to unfold one of the ubiquitin molecules in the polyubiquitin chain attached to the substrate and initiate its unfolding via insertion into the p97 central pore[3,16,17]. This affords a general mechanism for the recognition and processing of p97 substrates that is independent of the ubiquitylated target protein itself. The efficiency of p97-UFD1-NPL4-dependent unfolding reactions can be further stimulated by accessory cofactors. For instance, recent work showed that ubiquitin-binding UBX-type cofactors including FAF1 and UBXN7 stabilize p97-UFD1-NPL4 interactions with K48-linked polyubiquitin chains, involving multivalent interactions between these proteins and ubiquitin chains that promote the coordinated assembly of p97-cofactor complexes on polyubiquitylated proteins[18,19]. These and other studies have made it clear that multi-layered cofactor-driven mechanisms underlie the productive processing of ubiquitylated p97 client proteins, and partial functional redundancies between individual cofactors appear to be a common feature of the p97 system, ensuring its efficacy in maintaining protein homeostasis and cell fitness under both basal and stressful conditions. Whether additional as-yet unidentified p97 cofactors with important roles in these processes exist remains to be seen.

In this study, we discover a hitherto unrecognized vertebrate p97 cofactor named VCF1. We show that the interplay between VCF1 and p97 is mechanistically distinct from that of other cofactors, involving an α-helical motif in VCF1 that binds the p97 N-domain with unusually high affinity. VCF1 forms a joint complex with p97-UFD1-NPL4 and stimulates p97-UFD1-NPL4 interactions with ubiquitylated clients, promoting their turnover. However, VCF1 itself or in complex with p97 does not interact with ubiquitin, thus functioning as an unconventional accessory p97 cofactor that indirectly stimulates p97-UFD1-NPL4-dependent recognition and degradation of ubiquitylated target proteins via its tight binding to p97.

## Results

### VCF1 (FAM104A) is a p97-interacting protein

Through in silico searches for candidate regulators of p97 function, we came across FAM104A, a virtually uncharacterized protein detected in unbiased large-scale proteomic profiles of p97 interaction networks along with known p97 cofactors (Supplementary Fig. 1a)[20,21], but whose precise connection to p97 has not been studied. In humans, *FAM104A* encodes a relatively small protein of 186 amino acids that contains no annotated domains but is conserved across vertebrates (Supplementary Fig. 1b) and is broadly expressed, albeit at low levels, in human tissues. For reasons described in the following and in agreement with a recent study reporting similar findings on FAM104A[22], we refer to this protein as VCF1 (VCP/p97 nuclear Cofactor Family member 1). To investigate the function of VCF1, we generated cell lines expressing GFP-tagged VCF1 in a conditional manner, showing that GFP-VCF1 almost exclusively localizes to the nucleoplasm similar to ectopically expressed VCF1 containing a smaller Strep-HA tag (Fig. 1a, b; Supplementary Fig. 1c). The nuclear localization of VCF1 was gradually diminished upon progressive N-terminal truncations (Supplementary Fig. 1c), suggesting that VCF1 harbors several sequence determinants targeting it to the nucleus. Using cells stably expressing GFP-VCF1, we then performed mass spectrometry analysis of GFP pulldowns in order to profile the VCF1 interactome. Interestingly, this revealed a striking and highly selective enrichment of p97 (Fig. 1c; Supplementary Fig. 1d; Supplementary Data 1, 2). Notably, p97 efficiently co-precipitated with GFP-VCF1 under both mild and more stringent, partially denaturing RIPA buffer conditions (Fig. 1c; Supplementary Fig. 1d), suggesting that VCF1 forms a tight complex with p97. We generated a VCF1 antibody and used this to validate an interaction between endogenous VCF1 and p97 (Fig. 1d), supporting its physiological relevance. Moreover, purified recombinant VCF1 and p97 readily interacted in vitro (Fig. 1e; Supplementary Fig. 1e, f), demonstrating that the association between these proteins is direct. Reciprocal co-IP analysis showed that the *Xenopus laevis* VCF1 ortholog interacted with p97 in *Xenopus* egg extracts (Supplementary Fig. 1g), indicating that the VCF1-p97 interaction is evolutionarily conserved. Using purified full-length and truncated p97 proteins, we asked which of its domains is required for binding to recombinant VCF1. This revealed that VCF1 interacts with the p97 N-domain, similar to the majority of known p97 cofactors (Fig. 1f, g; Supplementary Fig. 1h)[11]. To further define the VCF1 binding site within the p97 N-domain, we analyzed the impact of mutating a range of individual residues in this domain that have been shown by structural and biochemical studies to be important for interaction with different p97 cofactors[23]. We found that Y143 in p97, which is critical for binding several cofactors[23,24], was required for binding to VCF1 (Fig. 1e), further confirming that VCF1 associates with the p97 N-domain. Having identified a p97 point mutation (Y143A) that abrogates VCF1 binding, we performed surface plasmon resonance (SPR) experiments to more quantitatively characterize the VCF1-p97 interaction. Strikingly, this showed that full-length VCF1 binds p97 with very high affinity ($K_d$ ~10 nM) (Fig. 1h), consistent with our VCF1 interactome studies above. In fact, the observed p97-binding affinity of VCF1 is substantially higher than those reported for other p97 cofactors, such as p47/NSFL1C ($K_d$ in the ~200−500 nM range)[25,26]. As expected, the p97 Y143A mutant did not show detectable binding affinity for VCF1 in SPR experiments (Fig. 1h). Together, these findings reveal VCF1 as a p97-binding protein that interacts tightly with the p97 N-domain.

## VCF1 binds p97 via a high-affinity motif that is distinct from known p97-binding determinants

We next asked how VCF1 associates with p97. Most p97 cofactors contain one or more well-defined p97-interacting motifs, including the VIM, VBM and SHP short linear motifs and the UBX/UBX-like domains[11]. However, no sequences conforming to any of the known p97-interacting motifs were present in VCF1. To identify the p97-binding region in VCF1, we therefore used a series of overlapping peptides spanning the full-length VCF1 protein as baits in p97-binding experiments (Fig. 2a). Incubation of these peptides with total cell extracts showed that the peptide encompassing the VCF1 C-terminus (VCF1-6) efficiently pulled down p97, whereas none of the other VCF1 peptides displayed detectable p97 binding (Fig. 2b). Interestingly, this C-terminal portion of VCF1 harboring the p97-binding determinant

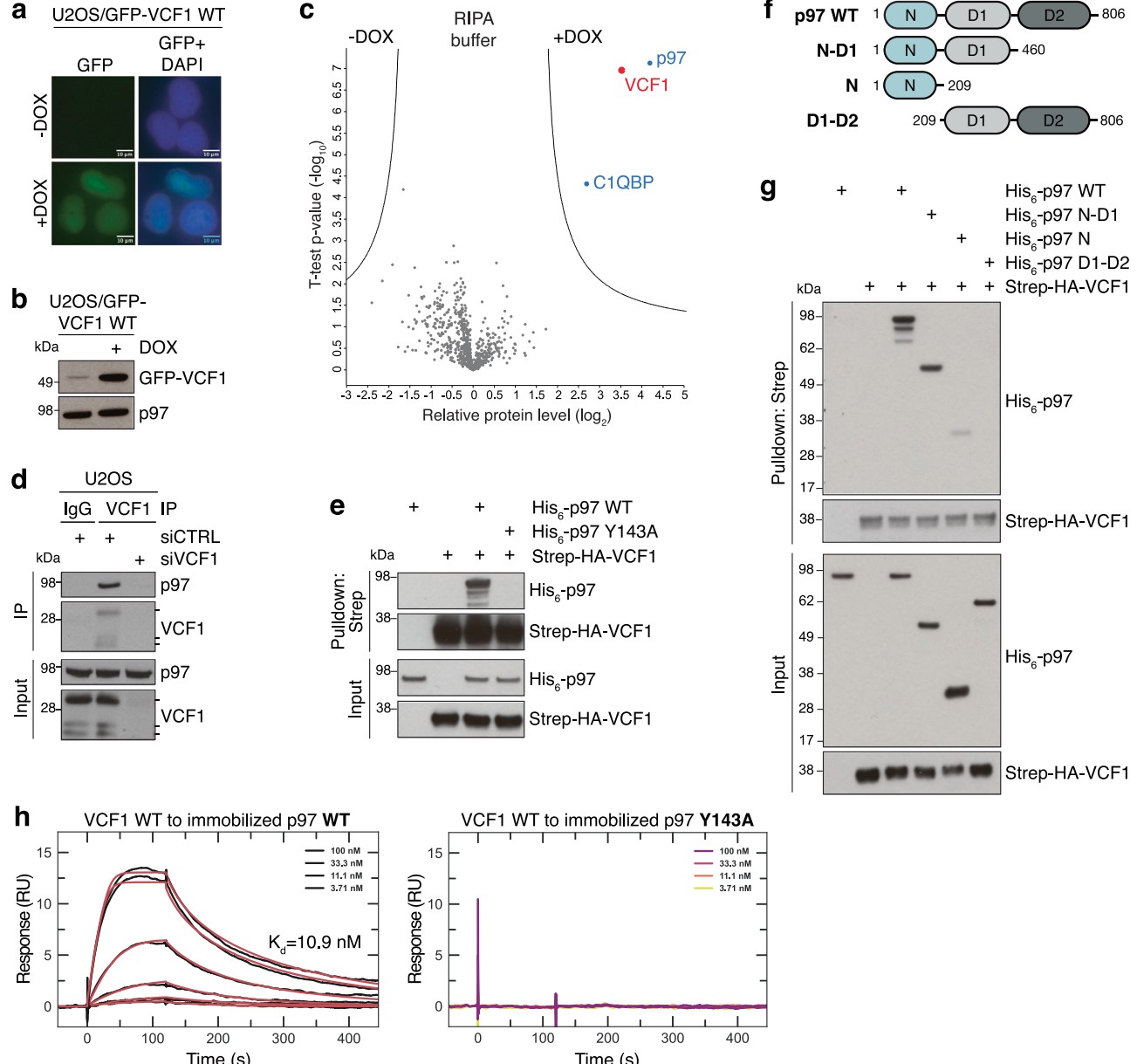

**Fig. 1 | VCF1 is a p97-binding protein that interacts tightly with the p97 N-domain. a** Representative images of U2OS/GFP-VCF1 WT cells treated or not with Doxycycline (DOX) for 16 h. Scale bars, 10 µM. **b** Immunoblot analysis of U2OS/GFP-VCF1 WT cells treated as in (**a**). **c** Mass spectrometry analysis of VCF1-interacting proteins. U2OS/GFP-VCF1 WT cells were treated or not with DOX for 16 h, subjected to GFP immunoprecipitation (IP) in partially denaturing RIPA buffer and analyzed by mass spectrometry. Volcano plot shows enrichment of individual proteins (+DOX/-DOX ratio) plotted against the *P* value (Supplementary Data 1). Dashed lines indicate the significance thresholds (two-sided *t* test, FDR < 0.05, $s_O = 1$). **d** Immunoblot analysis of VCF1 or pre-immune serum (IgG) IPs from whole cell lysates of U2OS cells transfected with non-targeting control (CTRL) or VCF1 siRNAs. Three specific bands corresponding to different VCF1 isoforms are detected by the VCF1 antibody, with the slower-migrating major band representing the 186-amino acid protein (isoform 1). **e** Immunoblot analysis of in vitro binding reactions in RIPA buffer containing purified His6-p97 and Strep-HA-VCF1 proteins that were subjected to StrepTactin (Strep) pulldown. **f** Schematic showing recombinant His6-p97 proteins used in (**g**). **g** As in (**e**), but using recombinant His6-p97 proteins shown in (**f**). Note that less His6-p97 N protein than His6-p97 WT and His6-p97 N-D1 co-purifies with Strep-HA-VCF1, since the isolated p97 N domain does not form hexamers, unlike p97 WT and p97 N-D1[2]. **h** Surface Plasmon Resonance (SPR) sensorgrams for the interaction between recombinant Strep-HA-VCF1 WT and immobilized His6-p97 proteins. For His6-p97 WT (left panel), black traces represent the experimental data and red traces correspond to the global fit of the two replicates according to a 1:1 interaction model. For the His6-p97 Y143A mutant (right panel), no dose-dependent response was observed. Data are representative of three (**a, b, d, e, g, h**) independent experiments with similar outcome. Source data are provided as a Source Data file.

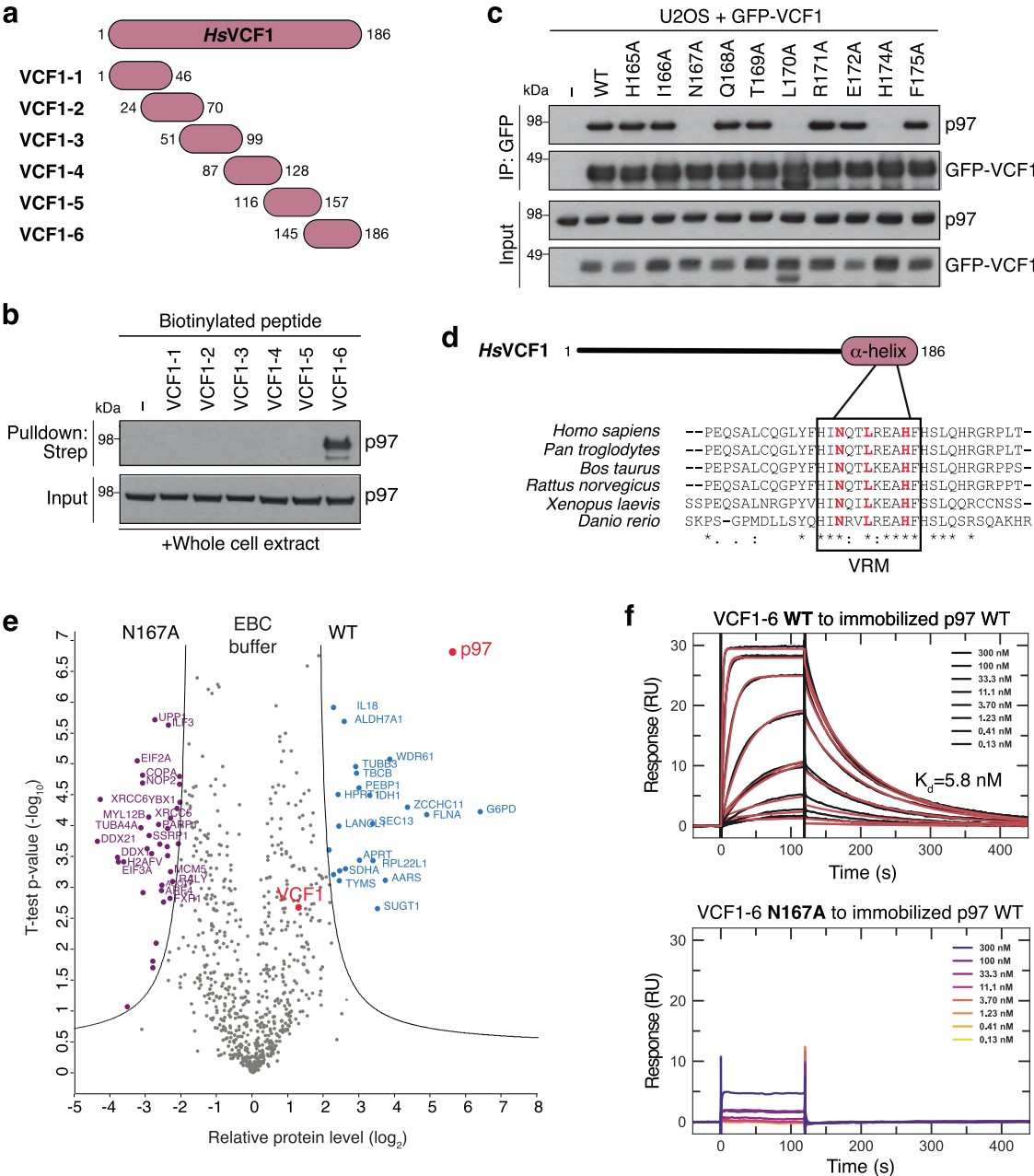

**Fig. 2 | VCF1 binds p97 via an α-helical motif (VRM). a** Schematic of VCF1 peptides used in (**b**). **b** Biotinylated VCF1 peptides in (**a**) were incubated with whole cell extracts of U2OS cells, subjected to StrepTactin (Strep) pulldown in RIPA buffer and analyzed by immunoblotting. **c** Immunoblot analysis of GFP IPs from U2OS cells transfected with indicated GFP-VCF1 expression constructs. **d** Sequence alignment showing conservation of the p97-binding motif (VRM) in selected vertebrate VCF1 proteins, with residues essential for p97 binding highlighted in red. **e** Comparison of VCF1 WT and N167A interactomes. U2OS/GFP-VCF1 WT or N167A cells treated with DOX for 16 h were subjected to GFP IP in non-denaturing EBC buffer and analyzed by mass spectrometry. Volcano plot shows enrichment of individual proteins (WT/N167A ratio) plotted against the $P$ value (Supplementary Data 3). Dashed lines indicate the significance thresholds (two-sided $t$ test, FDR < 0.01, $s_0 = 2$). **f** SPR sensorgrams for the interaction between VCF1-6 peptide (**a**) and immobilized recombinant His$_6$-p97. For the VCF1-6 WT peptide (left panel), black traces represent the experimental data and red traces correspond to the global fit of the two replicates according to a 1:1 interaction model. For the VCF1-6 N167A peptide (right panel), no dose-dependent response was observed. Data information: Data are representative of two (**b**, **c**, **f**) independent experiments with similar outcome. Source data are provided as a Source Data file.

represents its most highly conserved part and contains a region predicted by AlphaFold2[27] to adopt an α-helical fold, whereas the remaining VCF1 sequence is predicted to be mostly disordered (Supplementary Figs. 1b, 2a). To more precisely delineate the p97-binding motif in the VCF1 C-terminus, we performed an alanine scan of residues within the predicted α-helix in the context of full-length VCF1. This revealed that N167, L170 and H174 are each essential for VCF1 binding to p97 (Fig. 2c). Of note, these residues are all highly conserved among VCF1 orthologs and reside on the same face of the α-

helix (Fig. 2d; see also Fig. 3b). Mass spectrometry-based comparison of the interactomes of stably expressed GFP-VCF1 wild-type (WT) and N167A mutant proteins further demonstrated that the N167A mutation led to a strong loss of p97 binding (Fig. 2e; Supplementary Fig. 2b, c; Supplementary Data 3). Moreover, in vitro binding experiments with purified proteins confirmed that the introduction of the N167A mutation in both full-length VCF1 and the VCF1-6 peptide abrogated binding to p97 (Fig. 2f; Supplementary Fig. 2d, e). SPR experiments showed that the isolated VCF1 C-terminus binds p97 with a $K_d$ of approx. 6 nM,

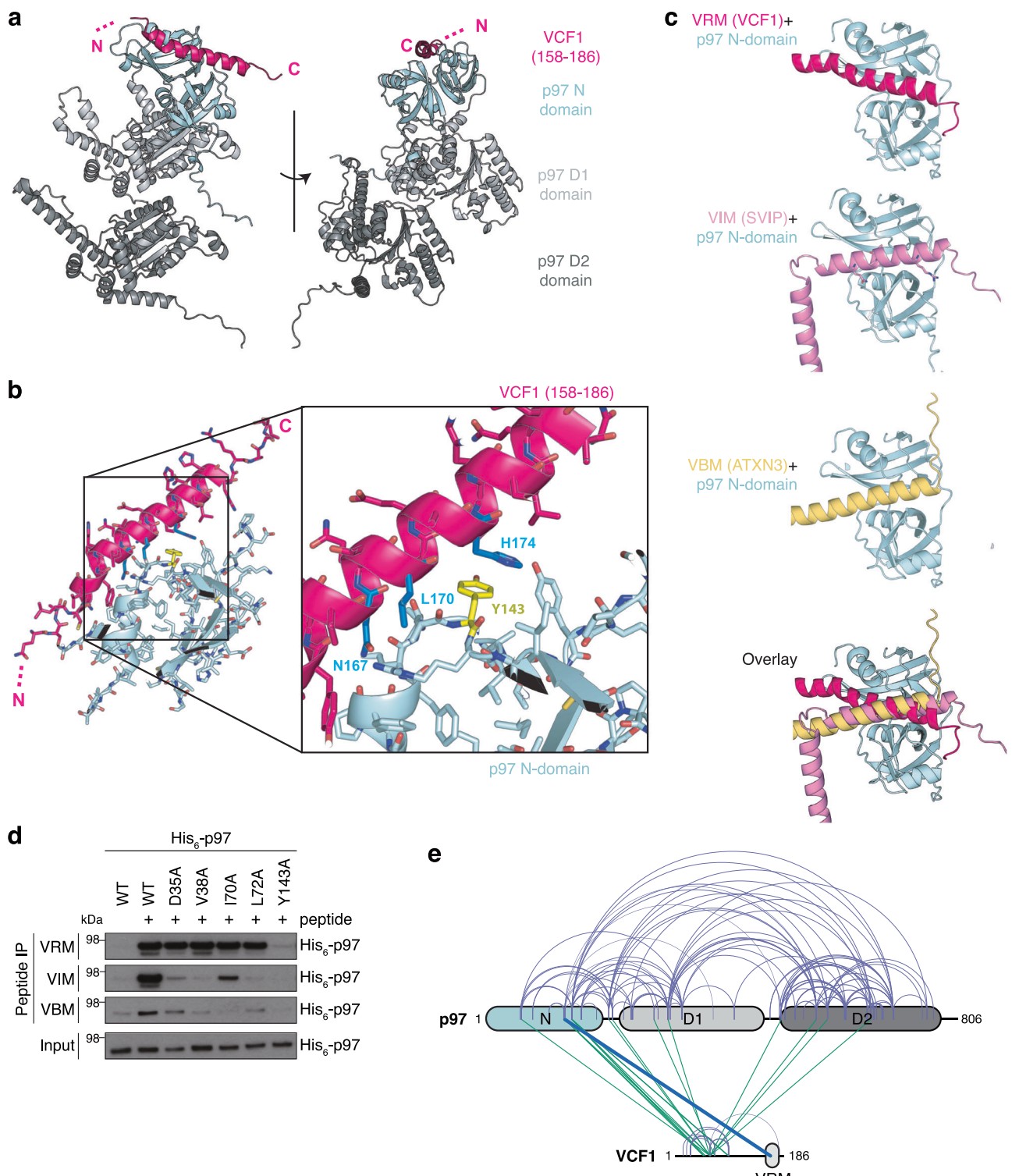

**Fig. 3 | Structural modeling of VCF1-p97 complex formation. a** AlphaFold-Multimer model of the complex between full-length VCF1 and a p97 monomer (colored by domains), predicting that the C-terminus of the otherwise highly unstructured VCF1 forms an α-helix (pink, residues 158–186) that interacts with the groove of the bipartite p97 N-domain (left panel). **b** Closeup view of the VCF1-p97 binding interface predicted by AlphaFold-Multimer (**a**), showing that it is involves Y143 (yellow) in the p97 N-domain and residues N167, L170 and H174 (blue) in VCF1 (right panel). **c** Comparison of AlphaFold-Multimer models of the complexes formed between the p97 monomer (N-domain in pale cyan) and VCF1 (pink), SVIP (rosa) or ATXN3 (yellow). **d** Biotinylated peptides spanning the VRM in VCF1, VIM in SVIP or VBM in ATXN3 immobilized on StrepTactin agarose were incubated with

purified WT or mutant His$_6$-p97 proteins, washed extensively in RIPA buffer and analyzed by immunoblotting. **e** Schematic visualization of all heteromeric VCF1-p97 crosslinks (green) identified by mass spectrometry analysis of purified FLAG-p97 and Strep-HA-VCF1 proteins incubated with the crosslinker disuccinimidyl dibutyric urea (DSBU) (Supplementary Fig. 3g). Self-crosslinks are shown in light purple. All visualized crosslinks scored >50 using MeroX, with a score of at least 19.4 required to ensure an FDR < 0.01 via decoy analysis. The line highlighted in blue visualizes a crosslink (with MeroX score of 116) between the p97 N-domain (K136) and the VCF1 VRM (Y163) (Supplementary Fig. 3f). Data information: Data are representative of two (**d**) independent experiments with similar outcome. Source data are provided as a Source Data file.

comparable to our results for full-length VCF1 (Figs. 2f, 1h). These data show that VCF1 contains a previously unrecognized type of linear motif mediating binding to p97 with unprecedented high affinity among p97 cofactors, which we refer to as the VRM (VCF1 p97-Recognizing Motif) (Fig. 2d).

## Structural modeling of the VCF1-p97 interaction

Having defined the primary sequence determinants in VCF1 and p97 underlying their association, we next sought to understand the structural basis of VCF1-p97 complex formation. Our attempts to obtain a cryo-EM structure of this complex were not successful, likely due to most of the VCF1 protein being disordered (Supplementary Fig. 2a), hence we used AlphaFold-Multimer[28] to model the VCF1-p97 interaction. This analysis revealed with high confidence that the VRM-containing α-helix in VCF1 docks at the interface between the two subdomains of the p97 N-domain involving N167, L170 and H174 in VCF1 and Y143 in p97 (Fig. 3a, b; Supplementary Fig. 3a, b), in excellent agreement with our experimental data. This mode of binding bears significant resemblance to that reported for other p97-interacting motifs, in particular the VIM and VBM that also form α-helices[11,29]. Indeed, comparison of the AlphaFold-Multimer models for p97 N-domain binding to the VRM and other p97-interacting motifs suggested that VRM interacts with the cleft between the two lobes of the p97 N-domain in a manner that is similar but not identical to the VIM and VBM motifs (Fig. 3c). The overlap between the N-domain surfaces recognized by the VRM and UBX domain is more limited, while the SHP box binds a distal region of the p97 N-domain that does not overlap with the VRM binding surface (Supplementary Fig. 3c)[11]. Consistent with both similarities and differences in the contacts made between p97 and the VRM, VIM and VBM motifs, Y143 in p97 was required for binding to all three motifs but not p47, whereas a number of other residues in the p97 N-domain known to make critical contacts with VIM and/or VBM[23] were dispensable for binding to VRM (Fig. 3d; Supplementary Fig. 3d). Of note, the AlphaFold-Multimer models predicted that the surface area buried by VRM on the p97 N-domain is considerably larger than the corresponding one for the VIM and VBM motifs (~1100 Å$^2$ for VRM and ~750 Å$^2$ for VIM and VBM) (Supplementary Fig. 3e), in good agreement with the higher p97 binding affinity of VCF1 relative to other known p97 cofactors. Interestingly, crosslinking mass spectrometry analysis with purified proteins showed that a number of contacts are formed between VCF1 sequences N-terminal to the VRM and p97 (Fig. 3e; Supplementary Fig. 3f, g), in accordance with the AlphaFold-Multimer model of the interaction between full-length VCF1 and p97 (Supplementary Fig. 3a). This suggests that in addition to high affinity p97 binding via the VRM a range of weak contacts involving the intrinsically disordered region of VCF1 may also contribute to the VCF1-p97 interplay. Together with our findings above, these data illuminate the molecular basis of the VCF1-p97 interaction.

## VCF1 and UFD1-NPL4 form a joint complex with p97

The notion that VCF1 and p97 cofactors containing VIM, VBM and UBX motifs recognize partially overlapping p97 N-domain surfaces suggests that they are mutually exclusive for binding to individual p97 protomers. However, VCF1 might form joint complexes with other cofactors on p97 hexamers. To address this possibility, we first used mass photometry to determine the stoichiometry of the VCF1-p97 interaction. This analysis suggested that, in the absence of other cofactors, 2-3 WT VCF1 molecules can bind a p97 hexamer (Fig. 4a), which may permit other cofactors to associate with the VCF1-p97 complex. We performed in vitro binding experiments with purified proteins to examine this possibility for several p97 cofactors including UFD1-NPL4, p47 and FAF1. Moderate concentrations of VCF1 were clearly permissive for the simultaneous binding of VCF1 and UFD1-NPL4, p47 or FAF1 to p97, as evidenced by the presence of VCF1 in

UFD1, p47 or FAF1 pulldowns that required its ability to bind p97 (Fig. 4b; Supplementary Fig. 4a–e). Likewise, substantiating the formation of a joint p97-VCF1-UFD1-NPL4 complex, we found that UFD1-NPL4 co-purified with immobilized VCF1 in a manner that was dependent on the presence of p97 and the p97-binding proficiency of VCF1 (Fig. 4c; Supplementary Fig. 4f). Gel filtration analysis provided further direct evidence for the formation of a stable complex between VCF1, UFD1-NPL4 and p97 (Fig. 4d), and NPL4 could be detected in endogenous VCF1 IPs along with p97 (Supplementary Fig. 4g; Supplementary Fig. 1f). Since our mass photometry analysis suggested a stoichiometry of 2-3 bound VCF1 molecules per p97 hexamer (Fig. 4a), we used AlphaFold-Multimer prediction to generate a model of a UFD1-NPL4-bound p97 hexamer that agrees with experimentally determined structures[30–32] and superimposed three VCF1 VRM-containing helices onto this model (Supplementary Fig. 4h, i). No steric clashes were observed (Supplementary Fig. 4h), indicating that several copies of VCF1 should be able to associate with the p97-UFD1-NPL4 complex. We noticed that high concentrations of WT VCF1, but not the N167A mutant, displaced p47 and UFD1-NPL4 from p97 in vitro, whereas increasing the concentration of p47 did not lead to dissociation of VCF1 from p97 (Supplementary Fig. 4b, j, k), consistent with the very high p97-binding affinity of VCF1. However, VCF1 is unlikely to displace UFD1-NPL4 or p47 from p97 under physiological conditions, as endogenous VCF1 is expressed at a low level in cells (~6000 copies/HeLa cell) and is approximately 100-fold less abundant than both UFD1-NPL4 and p47, and ~1000-fold less abundant than p97[33]. We conclude from these findings that VCF1 engages in complex formation with p97 together with other cofactors, including the primary p97 cofactor UFD1-NPL4.

## VCF1 stimulates p97-UFD1-NPL4 recruitment to ubiquitylated clients to facilitate their degradation

UFD1-NPL4 is a major p97 cofactor complex with a crucial role in promoting p97-dependent unfolding of K48-ubiquitylated substrates to enable their proteasomal degradation[3,16,17]. To interrogate the functional relevance of the observed complex formation between VCF1 and p97-UFD1-NPL4, we asked whether and how VCF1 status impacts the degradation of the well-established p97-UFD1-NPL4 model substrate Ub(G76V)-GFP that reflects the properties of an endogenous p97 target[34] in cells stably expressing this fusion protein. The level of Ub(G76V)-GFP is normally kept extremely low due to its constitutive poly-ubiquitylation and ensuing p97-UFD1-NPL4- and proteasome-dependent degradation but can be efficiently stabilized by inhibition of p97 or the proteasome (Supplementary Fig. 5a)[35,36]. Depletion of NPL4 increased Ub(G76V)-GFP abundance as expected, whereas knockdown of p47, which has not been implicated in Ub(G76V)-GFP degradation, had no effect (Fig. 5a, b; Supplementary Fig. 5b). Of note, we could only achieve partial knockdown of NPL4 (Supplementary Fig. 5b), likely due to its essential role in p97-dependent protein degradation and cell fitness[15,37]. Interestingly, knockdown of VCF1 by two independent siRNAs stabilized Ub(G76V)-GFP to a similar extent as reducing NPL4 levels (Fig. 5a, b; Supplementary Fig. 5b), suggesting that VCF1 is functionally involved in the p97-UFD1-NPL4-dependent processing of ubiquitylated proteins. By contrast, VCF1 depletion had a marginal impact on levels of a variant Ub(G76V)-GFP-20AA substrate that does not require p97 activity for its proteasomal degradation[35,38] (Supplementary Fig. 5a, c), consistent with VCF1 acting at the p97-dependent step of Ub(G76V)-GFP turnover. Likewise, VCF1 knockdown had no discernible effect on the p97-dependent processing of a Ub(G76V)-GFP protein rendered cytoplasmic by fusion to a nuclear export sequence (NES) (Ub(G76V)-GFP-NES; Supplementary Fig. 5d), suggesting that in line with its predominantly nuclear localization VCF1 mainly facilitates degradation of p97 client proteins in the nucleus. Remarkably, co-depletion of VCF1 and NPL4, but not p47, led to a dramatic accumulation of Ub(G76V)-GFP in the nucleus that greatly

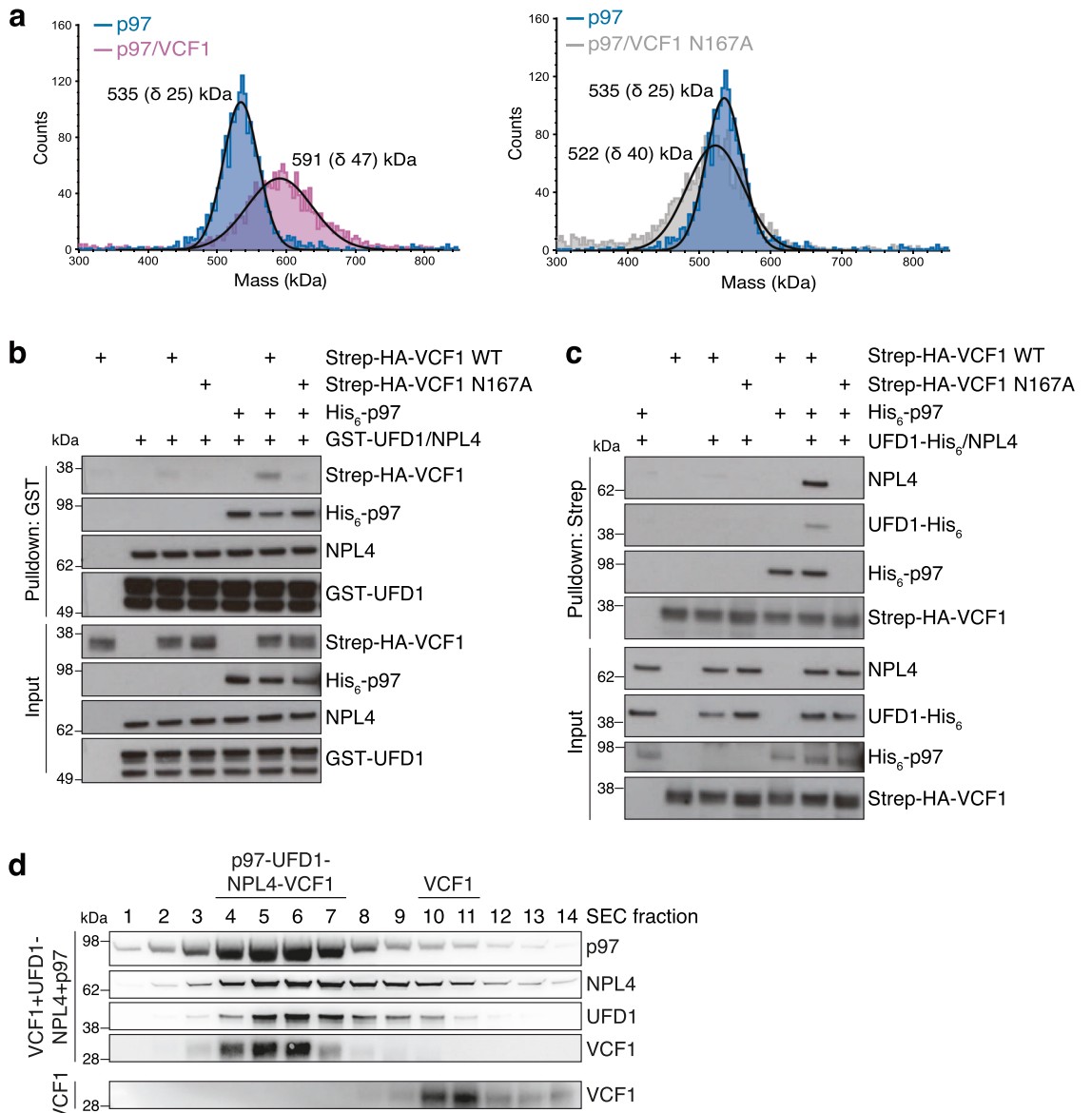

**Fig. 4 | VCF1 and UFD1-NPL4 form a joint complex with p97. a** Mass photometry-based mass distribution of His$_6$-p97 (blue) and His$_6$-p97 in the presence of saturating concentrations of full-length Strep-HA-VCF1 WT (magenta) (left panel) or N167A mutant (gray) (right panel). The mass photometry measurements show major species with molecular weights of 535 kDa (corresponding to a p97 hexamer) and 591 kDa (corresponding to a p97 hexamer bound by 2−3 VCF1 molecules). **b** Binding reactions containing indicated combinations of purified Strep-HA-VCF1, His$_6$-p97 and GST-UFD1-NPL4 proteins were subjected to Glutathione (GST) pulldown and analyzed by immunoblotting. **c** As in (**b**), except reactions containing indicated combinations of purified Strep-HA-VCF1, His$_6$-p97 and UFD1-His$_6$-NPL4 proteins were subjected to StrepTactin (Strep) pulldown. **d** Purified VCF1 was incubated alone or with equimolar concentrations of p97 and UFD1-NPL4 and then separated by size-exclusion chromatography (SEC). Immunoblot analysis of indicated fractions shows comigration of the p97-VCF1-UFD1-NPL4 complex. Data information: Data are representative of three (**b**, **c**) and two (**a**, **d**) independent experiments with similar outcome. Source data are provided as a Source Data file.

exceeded the impact of individual VCF1 or NPL4 knockdown (Fig. 5a, b; Supplementary Fig. 5b). Moreover, when Ub(G76V)-GFP was allowed to accumulate to high levels by treatment with a p97 inhibitor (p97i), depletion of VCF1 delayed the p97-dependent clearance of Ub(G76V)-GFP following p97i removal (Supplementary Fig. 5e). Thus, under conditions of reduced NPL4 availability or a high demand for p97 activity, VCF1 becomes crucial for p97-UFD1-NPL4-dependent turnover of ubiquitylated substrates. During unperturbed growth, however, VCF1 depletion only had a relatively modest impact on steady-state levels of ubiquitin conjugates and cell proliferation (Supplementary Fig. 5f, g). Overexpression of WT VCF1, but not the p97 binding-deficient N167A mutant, also stabilized Ub(G76V)-GFP similar to the effect of elevating NPL4 or p47 expression levels (Supplementary Fig. 5h), suggesting that a carefully controlled balance of p97

interactions with VCF1 and other cofactors is critical for the efficient processing of ubiquitylated substrates.

We next addressed how VCF1 stimulates p97-UFD1-NPL4-dependent protein degradation. The binding of VCF1 had no impact on the ATPase activity of p97 in vitro (Supplementary Fig. 6a). Moreover, the VCF1-p97 complex was unable to unfold a poly-ubiquitylated fluorescent model substrate (mEos3.2) of p97-UFD1-NPL4 in vitro (Supplementary Fig. 6b), suggesting that VCF1 does not promote an alternative, UFD1-NPL4-independent pathway for p97-mediated unfolding and degradation of ubiquitylated proteins. Prompted by recent work showing that UBX-type accessory p97 cofactors including FAF1 and UBXN7 stabilize p97-UFD1-NPL4 interactions with K48-linked ubiquitin chains[18,19], we considered the possibility that VCF1 may similarly enhance p97-UFD1-NPL4 binding to

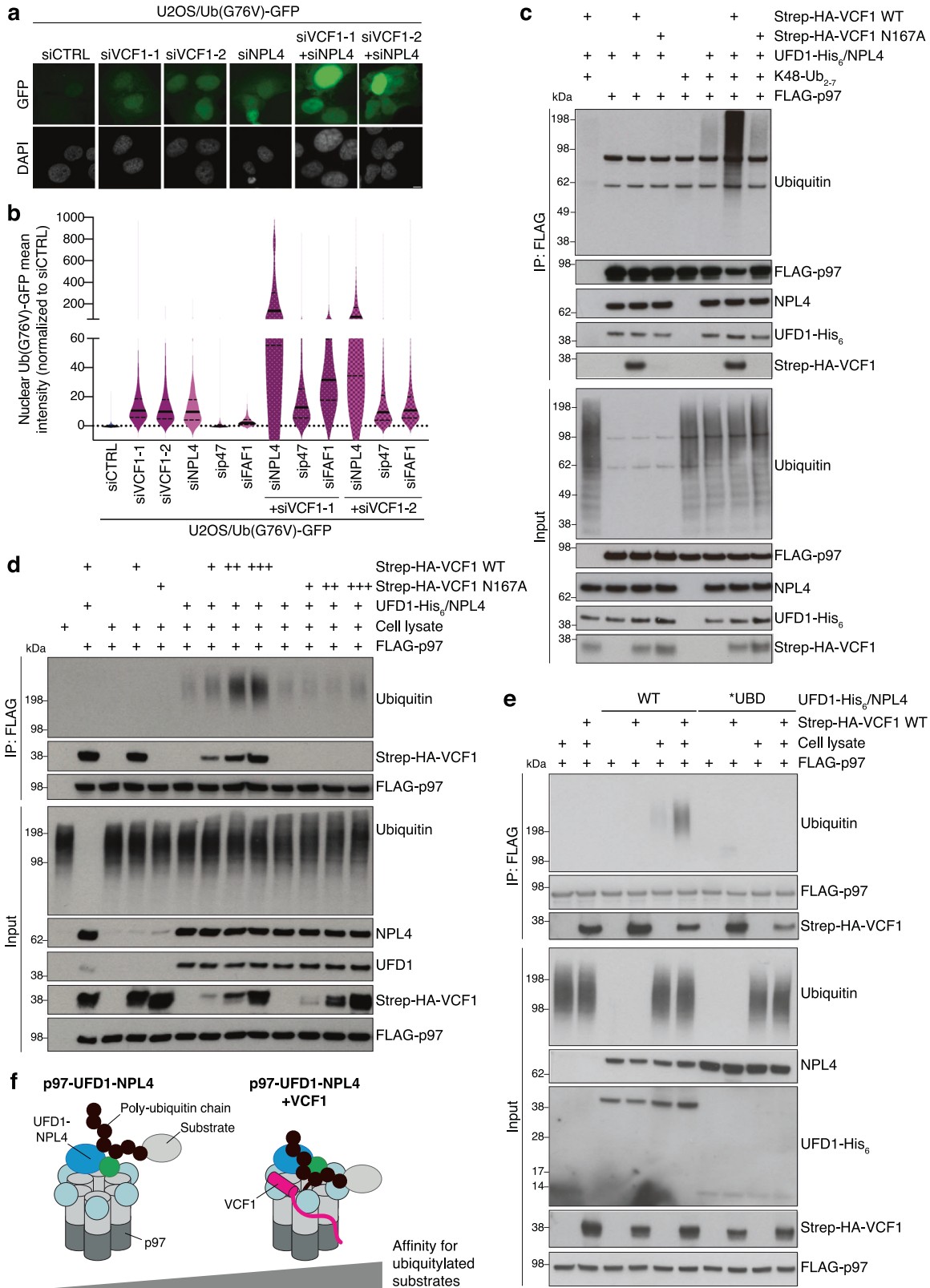

**Fig. 5 | VCF1 stimulates p97-UFD1-NPL4 recruitment to ubiquitylated substrates to facilitate their degradation. a** Representative images of U2OS/Ub(G76V)-GFP cells transfected with indicated siRNAs. Scale bar, 10 μM. **b** U2OS/Ub(G76V)-GFP cells transfected with indicated siRNAs were subjected to quantitative image-based cytometry (QIBC) analysis of Ub(G76V)-GFP expression (solid lines, median; dashed lines, quartiles; >10,000 cells analyzed per condition). **c** Immunoblot analysis of in vitro binding reactions containing indicated combinations of purified FLAG-p97, Strep-HA-VCF1, UFD1-His₆-NPL4 complex and K48-

linked ubiquitin chains (K48-Ub₂₋₇) that were subjected to FLAG IP. **d** As in (**c**), except that in vitro binding reactions containing purified proteins were incubated with whole cell extracts of U2OS cells where indicated and subjected to FLAG IP. **e** As in (**d**), using WT or ubiquitin binding-deficient (*UBD) UFD1-His₆-NPL4 complex. **f** Model of VCF1-dependent stimulation of p97-UFD1-NPL4 binding to ubiquitylated client proteins. See main text for details. Data information: Data are representative of three (**a**, **b**, **d**) and two (**c**, **e**) independent experiments with similar outcome. Source data are provided as a Source Data file.

ubiquitylated client proteins. Indeed, using purified proteins we found that WT VCF1 had a strong stimulatory impact on p97-UFD1-NPL4 binding to K48-linked poly-ubiquitin chains in vitro (Fig. 5c; Supplementary Fig. 6c). Substantiating this notion, we observed a similar prominent ability of VCF1 to enhance p97-UFD1-NPL4 binding to ubiquitylated proteins in total cell extracts (Fig. 5d). Importantly, this effect was abrogated by mutation of the VRM (Fig. 5c, d), demonstrating that the VCF1-dependent stabilization of complexes between p97-UFD1-NPL4 and ubiquitin chains requires its binding to p97. Strikingly, however, neither in isolation nor in complex with p97 did VCF1 display detectable interaction with ubiquitin unlike most p97 cofactors (Fig. 5d, e; Supplementary Fig. 6d). Moreover, VCF1 harbors no sequences conforming to known ubiquitin-binding domains, further arguing against a direct role of VCF1 in recruiting p97 to ubiquitylated substrates. Using truncated VCF1 proteins, we found that the C-terminal portion harboring the VRM was sufficient to enhance p97-UFD1-NPL4 interactions with ubiquitin conjugates (Supplementary Fig. 6e), suggesting that the binding of VCF1 to p97 per se represents a primary determinant for its stimulatory impact on p97-UFD1-NPL4 binding to ubiquitylated substrates. Consistent with this scenario, VCF1 did not directly interact with UFD1-NPL4 and had no discernible impact on UFD1-NPL4 association with p97 (Figs. 4b, c, 5c). Importantly, using ubiquitin binding-deficient UFD1 and NPL4 mutants[18,39,40] we established that the ability of VCF1 to boost p97-UFD1-NPL4 interactions with ubiquitylated proteins is fully dependent on the ubiquitin-binding ability of the UFD1-NPL4 complex (Fig. 5e), further demonstrating that the VCF1-p97 complex has no overt ubiquitin-binding ability on its own. VCF1 did not accelerate the already highly efficient p97-UFD1-NPL4-mediated unfolding of poly-ubiquitylated mEos3.2 in vitro (Supplementary Fig. 6b), possibly reflecting that this represents an optimal setting for p97-UFD1-NPL4-dependent binding to and unfolding of a poly-ubiquitylated substrate in vitro using purified components where physiologically relevant supporting roles of accessory cofactors could be masked. Together, the data suggest that VCF1 promotes p97-dependent protein degradation by stimulating the interaction of p97-UFD1-NPL4 with ubiquitylated substrates, via an unconventional mechanism differing from that of UBX-type cofactors by being principally mediated by the direct binding of VCF1 to p97 but not ubiquitin (Fig. 5f).

## Discussion

p97 is a nexus of the ubiquitin system, unfolding numerous ubiquitylated client proteins recruited by a broad range of cofactors. Here, we discovered that the small uncharacterized protein VCF1 defines a hitherto unrecognized p97 cofactor, displaying several unique features that set it apart from known cofactors. Our detailed biochemical and biophysical analyses revealed that VCF1 directly binds p97 with unusually high affinity relative to other cofactors, and that this is mediated by an α-helical motif (VRM) distinct from known p97-binding determinants that forms extensive contacts with the p97 N-domain. As VCF1 is a low-abundant protein whose copy number in HeLa cells is approximately 1000-fold lower than p97[33], it is likely that a large fraction of the endogenous VCF1 pool is constitutively associated with p97. Indeed, despite its low abundance the binding of VCF1 to p97 could be readily observed at the endogenous level, supporting the physiological importance of this interaction. Although VCF1 binds p97 with very high affinity by recognizing a surface on the p97 N-domain that partially overlaps with that of other p97-binding motifs, the VCF1-p97 complex is permissive for the simultaneous binding of additional N-domain-binding cofactors, as only 2-3 VCF1 molecules associate with a p97 hexamer even under saturating concentrations. We established that VCF1 forms a joint complex with UFD1-NPL4 on p97, and that this is functionally important for promoting p97-UFD1-NPL4-dependent turnover of a ubiquitylated model substrate in cells. Mechanistically,

we show that VCF1 acts in an unorthodox way to stimulate p97-UFD1-NPL4 binding to poly-ubiquitylated proteins via the ubiquitin-binding motifs of UFD1 and NPL4. In terms of the functional outcome, this mechanism of action shares features with that recently described for ubiquitin-binding UBX-type p97 cofactors including FAF1 and UBXN7, which stabilize p97-UFD1-NPL4 interactions with K48-linked poly-ubiquitin chains to promote ubiquitin-dependent unloading of the replicative CMG helicase from chromatin[18,19]. Thus, like these UBX-type cofactors, VCF1 promotes the targeting of p97 to ubiquitylated client proteins in a substrate-agnostic manner by enhancing the affinity of p97-UFD1-NPL4 for poly-ubiquitin modifications (Fig. 5f). However, contrary to the UBX proteins, VCF1 does not possess detectable intrinsic ubiquitin-binding ability but instead augments the ubiquitin-binding affinity of the p97-bound UFD1-NPL4 complex indirectly. We found that this is primarily mediated by the docking of the VRM on the p97 N-domain, which is necessary and sufficient for enhancing the ubiquitin-binding capacity of the p97-UFD1-NPL4 complex. In line with the VRM providing a principal contribution to stimulating p97-UFD1-NPL4 interactions with ubiquitin chains, most of the VCF1 sequence apart from the VRM-containing α-helix is relatively poorly conserved among vertebrates (Supplementary Fig. 1b). However, using cross-linking mass spectrometry we found that the intrinsically disordered portion of VCF1 also makes a number of contacts to p97. This suggests that in addition to the tight VRM-p97 N-domain association multivalent low-affinity VCF1-p97 interactions involving sequences outside the VRM may also be relevant for the role of VCF1 in regulating p97 functions in cells, analogous to previous findings for disordered regions within the p97 cofactor UBXD1[41]. Elucidating the precise molecular basis of how VCF1 binding to the p97 N-domain impacts the configuration of the p97-UFD1-NPL4 complex to enhance its interaction with ubiquitin chains will ultimately require structural insights, which may be challenging given the intrinsically disordered nature of most of the VCF1 protein. Interestingly, a recent study independently identified VCF1 as a p97-binding protein associating with the p97-UFD1-NPL4 complex, reporting a role of VCF1 in promoting nuclear import of p97[22]. Thus, VCF1 may stimulate processing of nuclear p97-UFD1-NPL4 client proteins via several mechanisms.

The complementary mechanisms of action of VCF1 and UBX-type cofactors in enhancing p97-UFD1-NPL4 interactions with ubiquitylated target proteins may be an important component of multi-layered processes ensuring the robustness and flexibility of the p97 system in unfolding large swathes of ubiquitylated client proteins throughout the cell. We found that when the availability of p97-UFD1-NPL4 is limited by partial depletion of NPL4, VCF1 becomes instrumental for turning over the p97-UFD1-NPL4 model substrate Ub(G76V)-GFP. Moreover, high levels of Ub(G76V)-GFP requires VCF1 function for efficient p97-dependent clearance. During unperturbed conditions, however, depletion of VCF1 has little impact on overall levels of ubiquitin conjugates and cell fitness. This suggests that VCF1 and other cofactors may have at least partially redundant roles in stimulating p97-UFD1-NPL4 interactions with and processing of ubiquitylated targets under steady-state conditions where the basal activity of this complex is not rate-limiting, whereas individual cofactors may become more critical when demand for p97-UFD1-NPL4-mediated protein unfolding and degradation surges. Whether VCF1 acts as a non-selective enhancer of p97-UFD1-NPL4-dependent protein turnover, as might be suggested by the apparent lack of domains other than its conserved p97-binding motif, or promotes p97-UFD1-NPL4 function in specific contexts remains to be established. However, given its high affinity for p97 binding it seems likely that VCF1 is a preferred accessory cofactor for enhancing the basal ubiquitin-binding capacity of the p97-UFD1-NPL4 complex. Indeed, VCF1 depletion had a much stronger impact on Ub(G76V)-GFP levels than FAF1 knockdown. While we cannot rule out that the additive impact of VCF1 knockdown and partial NPL4 depletion in stabilizing Ub(G76V)-GFP could reflect that VCF1 and UFD1-NPL4 act

in independent pathways of p97-dependent protein degradation, we consider this unlikely since the VCF1-p97 complex on its own does not associate with and unfold ubiquitylated substrates in vitro, and degradation of the Ub(G76V)-GFP model substrate is not known to be dependent on primary cofactors other than UFD1-NPL4. Irrespective of the precise mechanism, our studies show that the ratio between VCF1 and other p97 cofactors must be carefully balanced to ensure proper functionality of p97-UFD1-NPL4-dependent protein degradation. The tight p97 binding of VCF1 could make it pertinent for its expression levels to be kept low to avoid disrupting such equilibria. At present, the mechanisms controlling p97-cofactor interactions are not well understood but are likely subject to a considerable degree of regulatory complexity given the multitude of p97 functions and cofactors. Future studies of the biological role of VCF1 may shed important further light on the workings and regulation of the p97 system.

## Methods

### Cell culture

U2OS (catalog no. HTB-96) cells were obtained from ATCC. HEK293-EBNA1-6E cells were a kind gift from Yves Durocher (National Research Council Canada, Montreal, Canada). All mammalian cell lines were cultured under standard conditions at 37 °C and 5% $CO_2$ in DMEM (Thermo) supplemented with 10% FBS (v/v) and penicillin-streptomycin (Thermo). Recombination cloning-compatible U2OS cells were generated using the Flp-In T-Rex Core Kit (Thermo) according to the manufacturer's instructions and maintained under selection using Blasticidin (Invivogen) and Zeocin (Thermo). To generate cell lines stably expressing GFP-VCF1, U2OS Flp-In T-Rex cells (Thermo) were co-transfected with GFP-VCF1 constructs and pOG44 (Thermo) plasmids, followed by selection with Hygromycin (Invitrogen). Single clones were carefully screened for uniform expression of ectopic GFP-VCF1 at low levels and maintained under selection with Blasticidin and Hygromycin. To induce GFP-VCF1 expression, cells were treated with Doxycycline (DOX) (0.5 μg/ml) for 16 h. U2OS cell lines stably expressing Ub(G76V)-GFP or Ub(G76V)-GFP-20AA were described previously[42]. A U2OS cell line stably expressing Ub(G76V)-GFP-NES was generated under G418 (Invivogen) selection. Clones with low baseline GFP expression were visually screened for low basal GFP expression. All cell lines used in this study were regularly tested negative for mycoplasma infection and were not authenticated.

Relative cell proliferation was measured using an Incucyte S3 Live-Cell Analysis System. Cells were treated with siRNAs 36 h prior to seeding. Cells ($2 \times 10^4$) were seeded in triplicate into a 12-well plate. Cells were imaged at 6-h intervals with a mean confluency determined from 9 images per well.

Unless otherwise indicated, the following drug concentrations were used: NMS-873 (p97i; 5 μM, Sigma), MG132 (20 μM; Sigma).

### Xenopus egg extracts

Nucleoplasmic egg extracts (NPE) were prepared using *Xenopus laevis* (LM0053MX, LM00715MX, Nasco). All experiments involving animals were approved by the Danish Animal Experiments Inspectorate and conform to relevant regulatory standards and European guidelines. Preparation of *Xenopus* egg extracts was performed as described previously[43,44].

### Plasmids and siRNAs

Bacterial expression constructs for $His_6$-tagged p97 fragments were generated by Q5 Site-Directed mutagenesis (New England Biolabs) of pET15b-$His_6$-p97 plasmid, using the following primers: 5′-TGACGATATCATGGCTTCTGGAGCC-3′ and 5′-TGACGCGGCCGCCTAGCCATACAGGTCA-3′ (for full-length p97); 5′-TGACGCGGCCGCCTATTCGCAGTGGATCACTGTG-3′ (for p97 N); 5′-TGACGGCCGGCCGCCTAGCCCCCGATGTCTTC-3′ (for p97 N + D1); 5′-TGACGATATCAAGCAGCTAGCTC

AG-3′ (for p97 D1 + D2). Mammalian expression constructs for GFP-VCF1 WT and Strep-HA-VCF1 were generated by PCR amplification of a synthetic *VCF1* cDNA isoform (Thermo Fisher Scientific) encoding isoform 1 of human FAM104A (UniProt ID: Q969W3) and insertion into pcDNA5/FRT/TO/EGFP or pcDNA5/FRT/TO/Strep-HA (Thermo), respectively, using KpnI and ApaI. Mammalian expression construct for FLAG-VCF1 was generated by inserting *VCF1* cDNA into p-FLAG-CMV-2 (Sigma). Point mutations and deletions were introduced using the Q5 Site-Directed Mutagenesis kit (New England Biolabs). pET41b + _Ufd1-$His_6$ (#117107), pET41b + _Npl4 (#117108) and pcDNA5/FRT/TO/FAF1-Strep-HA (#113486) were acquired from Addgene. Ubiquitin binding-deficient (*UBD) mutants of UFD1-$His_6$ (deletion of amino acids 2-215) and NPL4 (T590L + F591V point mutations)[18] were generated using the Q5 Site-Directed Mutagenesis kit (New England Biolabs). Ub(G76V)-GFP-NES plasmid was generated from Ub(G76V)-GFP (Addgene, #11941) using the Q5 Site-Directed Mutagenesis kit and the primer set 5′-ACGCCTGACTCTGGATTAAAGCGGCCGCGACTCT-3′ and 5′-TCCAGAGGAGGCAGCTGCTTGTACAGCTCGTCCATGC-3′.

All siRNAs (Sigma) were used at 20 nM and transfected with Lipofectamine RNAiMAX reagent (Thermo) according to manufacturer's instructions. The following siRNA oligonucleotides were used: Non-targeting control (CTRL): 5′-GGGAUACCUAGACGUUCUA-3′; VCF1(#1): 5′-GACAGUGGUGGGAGCAGCA-3′; VCF1(#2): 5′-GUCUUCAGGCAGUGACAGU-3′; p47: 5′-AGCCAGCUCUUCCAUCUUA-3′; NPL4: 5′-CACGCCUAAUCCUGUUGACAA-3′; FAF1: 5′-CCACCUUCAUCAUCUAGUC-3′; p97: Dharmacon SmartPool (D-008727).

### Expression and purification of recombinant proteins

$His_6$-p97, $His_6$-p47 and FLAG-p97 proteins were expressed in chemically competent *E. coli* BL21(DE3) (Thermo Fisher Scientific) by adding 1 mM IPTG (AppliChem) for 3 h at 37 °C. Bacterial pellets were then lysed in low salt buffer (50 mM Tris-HCl, pH 7.5; 0.2 M NaCl; 10% Glycerol; 10 mM Imidazole; 0.1% Triton-X100; 2 mM DTT) containing protease inhibitors, and lysates were sonicated and cleared by centrifugation. Cleared lysates were incubated with equilibrated Ni-NTA beads (Qiagen), incubated for 2 h at 4 °C, washed in a gravity column using 100 ml high salt buffer (50 mM Tris-HCl, pH 7.5; 1 M NaCl; 10% Glycerol; 15 mM Imidazole; 0.1% Triton-X100; 2 mM DTT) and 75 ml low salt buffer, followed by elution using low salt buffer containing 350 mM imidazole ($His_6$-p97 and $His_6$-p47), or incubated with equilibrated FLAG beads (anti-FLAG M2 Affinity Gel, Sigma Aldrich), incubated for 2 h at 4 °C washed in a gravity column using 150 ml lysis buffer, followed by elution using lysis buffer containing 0.5 mg/ml 3X FLAG Peptide (Sigma Aldrich) (FLAG-p97). The elute fractions were run on a 4–12% NuPAGE Bis-Tris protein gel (Invitrogen) and stained with Instant Blue Coomassie Protein Stain (Expedeon). Fractions were concentrated on Microcon-30 kDa Centrifugal Filters (Millipore), pooled and loaded onto a Superose 6 10/300 GL column equilibrated in gel filtration buffer (20 mM HEPES, pH 7.5; 150 mM NaCl; 2 mM DTT). Fractions containing $His_6$-p97, $His_6$-p47 or FLAG-p97 were then pooled, concentrated, aliquoted and snap-frozen in liquid nitrogen.

For expression and purification of Strep-HA-VCF1 proteins, HEK293-EBNA1-6E suspension cells were cultured in FreeStyle F17 expression medium (Gibco) supplemented with 4 mM L-glutamine (Gibco), 1% FBS, 0,1% Pluronic F-68 (Gibco) and 50 μg/ml G418 (Invivogen) in a 37 °C incubator on a shaker rotating at 140 rpm. Cells were transfected with Strep-HA-VCF1 expression plasmids using PEI transfection reagent (Polyscience Inc). After 30 h, cells were harvested and snap-frozen in liquid nitrogen. The cell pellet was resuspended in lysis buffer (100 mM Tris, pH 8.0; 350 mM NaCl; 0,05% NP-40 and protease inhibitors), treated with benzonase and sonicated. The lysate was cleared by centrifugation at 25,000 g at 4 °C for 30 min and incubated with Strep-Tactin Superflow resin (IBA) at 4 °C for 2 h. Beads were washed in a gravity column using wash buffer (100 mM Tris pH 8.0;

600 mM NaCl; 0,05% NP-40) and Strep-HA-VCF1 protein was eluted using elution buffer (100 mM Tris pH 8.0; 350 mM NaCl; 2.5 mM Desthiobiotin (IBA)). The elute fractions were run on a 4–12% NuPAGE Bis-Tris protein gel (Invitrogen) and stained with Instant Blue Coomassie Protein Stain (Expedeon). Fractions were concentrated on Microcon-10 kDa Centrifugal Filters (Millipore), pooled and loaded onto a Superose 6 10/300 GL column equilibrated in gel filtration buffer (20 mM HEPES, pH 7.5; 150 mM NaCl; 2 mM DTT). Fractions containing VCF1 were then pooled, concentrated, aliquoted and snap-frozen in liquid nitrogen.

UFD1-His$_6$-NPL4 complex was expressed and purified as described previously[19]. Briefly, NPL4 and UFD1-His$_6$ proteins were expressed individually (ratio 2:1) overnight at 18 °C in chemically competent *E. coli* BL21(DE3) (Thermo Fisher Scientific) by addition of 1 mM IPTG (AppliChem). Bacterial pellets were resuspended in lysis buffer (50 mM Tris-HCl, pH 8.0; 0.5 M NaCl; 5 mM Mg(OAc)$_2$; 0.5 mM TCEP; 30 mM imidazole) containing protease inhibitors. The samples were then mixed, supplemented with lysozyme (Fisher Scientific) and sonicated before clearing the lysates by centrifugation. The supernatant was mixed with Ni-NTA beads (Qiagen), incubated for 2 h at 4 °C and the beads washed in a gravity column using 150 ml lysis buffer, followed by 10 ml of lysis buffer containing 50 mM imidazole, 2 mM ATP and 10 mM Mg(OAc)$_2$. The protein complex was then eluted in lysis buffer containing 400 mM imidazole. The elute fractions were run on a 4–12% NuPAGE Bis-Tris protein gel (Invitrogen) and stained with Instant Blue Coomassie Protein Stain (Expedeon). Fractions were concentrated on Microcon-10 kDa Centrifugal Filters (Millipore), pooled and loaded onto a Superose 6 10/300 GL column equilibrated in UFD1-NPL4 gel filtration buffer (20 mM HEPES-KOH, pH 7.9; 150 mM NaCl; 5 mM Mg(OAc)$_2$; 0.3 M sorbitol; 0.5 mM TCEP). Fractions containing the UFD1-His$_6$-NPL4 complex were then pooled, concentrated, aliquoted and snap-frozen in liquid nitrogen.

For purification of NPL4-GST-UFD1 complex, NPL4 and GST-UFD1 were expressed individually (ratio 2:1) in chemically competent *E. coli* BL21(DE3) (Thermo Fisher Scientific) by addition of 1 mM IPTG (AppliChem) for 3 h at 37 °C. Bacterial pellets were then mixed and lysed in lysis buffer (50 mM HEPES-KOH, pH 7.5; 150 mM KCl; 5% Glycerol; 2 mM MgCl$_2$; 0.5 mM TCEP) containing protease inhibitors, treated with lysozyme (Fisher Scientific), and lysates were sonicated and cleared by centrifugation. The cleared lysates were incubated with equilibrated Glutathione agarose (Gold Biotechnology) for 2 h at 4 °C and the beads washed in a gravity column using 50 ml lysis buffer, 50 ml high salt buffer (lysis buffer containing 300 mM KCl) and 50 ml lysis buffer. Proteins were eluted by adding lysis buffer containing 35 mM reduced glutathione (Gold Biotechnology). The elute fractions were run on a 4–12% NuPAGE Bis-Tris protein gel (Invitrogen) and stained with Instant Blue Coomassie Protein Stain (Expedeon). Fractions were concentrated on Microcon-30 kDa Centrifugal Filters (Millipore), pooled and loaded onto a Superose 6 10/300 GL column equilibrated in NPL4-GST-UFD1 gel filtration buffer (50 mM HEPES pH 7.5; 150 mM KCl; 2 mM MgCl$_2$; 0.5 mM TCEP). Fractions containing the NPL4-GST-UFD1 complex were then pooled, concentrated, aliquoted and snap-frozen in liquid nitrogen.

### Antibodies
The following commercial antibodies were used: FLAG (A00187, GenScript, mouse, 1:1000 (RRID:AB_1720813)); HA (11867423001, Roche, rat, 1:1000 (RRID:AB_390918)); HA (sc-7392, Santa Cruz, mouse, 1:1000 (RRID:AB_627809)); HA (sc-805, Santa Cruz, rabbit, 1:1000 (RRID:AB_631618)); HA (MMS-101R, Covance, mouse, 1:1000 (RRID:AB_291262)); His$_6$ (631212, Clontech, mouse, 1:500 (RRID:AB_2721905)); NPL4 (sc-365796, Santa Cruz, mouse, 1:500 (RRID:AB_10841920)); NPL4 (HPA021560, Atlas Antibodies, rabbit, 1:1000 (RRID:AB_1854586)); (UFD1 (sc-136114, Santa Cruz, mouse, 1:500 (RRID:AB_2213950)); UFD1 (611642, BD Biosciences, mouse,

1:1000 (RRID:AB_399070)); Ubiquitin (43124S, Cell Signaling Technology, rabbit, 1:1000 (RRID:AB_2799235)); Ubiquitin (sc-8017, Santa Cruz, mouse, 1:1000 (RRID:AB_628423)); p97 (MA3-004, Invitrogen, mouse, 1:5.000 (RRID:AB_2214638)); p97: Santa Cruz, mouse, sc-57492 (RRID:AB_793927)); GST (sc-138, Santa Cruz, mouse, 1:1000 (RRID:AB_627677)); p47 (NBP213677, Novus Biologicals, rabbit, 1:1000). A sheep polyclonal VCF1 antibody was raised against full-length human VCF1 by MRC PPU Reagents and Services (University of Dundee, UK) and used at a 1:200 dilution.

Antibodies against *Xenopus* p97, NPL4 and UFD1[45] were a kind gift from Olaf Stemmann (University of Bayreuth, Germany). The following antibodies against *Xenopus* proteins were raised against the indicated peptides (New England Peptide): VCF1 N-terminus (H2N-MLPDSRKRRRNSNEVSEC-amide), VCF1 C-terminus (Aoa-KEAHFSSLQQRCCNSS-OH). The VCF1 antibody used for immunoblotting was raised against full-length *Xenopus laevis* VCF1 (Biogenes). The VCF1 protein was tagged with His$_6$ on the N-terminus and purified under denaturing conditions.

### Cell lysis, immunoblotting and immunoprecipitation
Immunoblotting was performed as previously described[46]. Immunoprecipitation (IP) of HA-tagged proteins was performed using Anti-HA Affinity matrix (Roche) under denaturing conditions, unless otherwise stated. GFP-trap and DYKDDDDK Fab-trap agarose (Chromotek) were used according to manufacturer's instructions. Cell lysis was performed using RIPA buffer (140 mM NaCl; 10 mM Tris-HCl, pH 8.0; 0.1% sodium deoxycholate; 1% Triton X-100; 0.1% SDS; 1 mM EDTA; 0.5 mM EGTA) or EBC buffer (50 mM Tris, pH 7.5; 150 mM NaCl; 1 mM EDTA; 0.5% NP-40, 1 mM DTT). IPs were washed in non-denaturing buffer (50 mM Tris-HCl, pH 7.5; 150 mM NaCl; 0.5 mM EDTA). Cell lysis and IP under partially denaturing conditions were performed using denaturing buffer (20 mM Tris-HCl, pH 7.5; 50 mM NaCl; 0.5% sodium deoxycholate; 0.5% NP-40; 0.5% SDS; 1 mM EDTA). All lysis and wash buffers were supplemented with 1 mM fresh PMSF Protease Inhibitor (Thermo), complete EDTA-free protease inhibitor Cocktail Tablets (Roche), 1.25 mM N-ethylmaleimide (Sigma) and 50 μM PR-619 (Calbiochem).

To immunodeplete VCF1, p97 and NPL4 from *Xenopus* egg extracts, one volume of Protein A Sepharose beads (GE Healthcare) was mixed with two volumes of VCF1 N-terminus (1 mg/ml), VCF1 C-terminus (1 mg/ml), p97 and NPL4 antibodies and incubated overnight at 4 °C. After incubation, beads were washed 4 times with ELB buffer (10 mM HEPES, pH 7.7; 50 mM KCl; 2.5 mM MgCl$_2$; 250 mM sucrose). One volume of NPE was mixed with one volume of beads for 1 h at room temperature. Supernatant samples were harvested and beads were washed 4 times using ELB buffer containing 0,25% NP-40 and resuspended in 2x Laemmli sample buffer.

### Immunofluorescence
Mammalian cells grown on coverslips were fixed in formalin buffer (VWR) for 15 min at room temperature (RT). Cells were permeabilized with PBS containing 0.5% Triton X-100 for 5 min and blocked with 5% BSA (Sigma) for 1 h prior to staining with indicated primary antibodies for 1 h at RT. After 3 washes cells were stained with a combination of Alexa Fluor secondary antibodies and 4′,6-Diamidino-2-Phenylindole (DAPI; Molecular Probes) for 1 h at room temperature in the dark. Finally, after 4 further rounds of washing, coverslips were dried and mounted on glass slides using Mowiol (Sigma). Quantitative image-based cytometry (QIBC) was performed as described[47]. In brief, cells were fixed, permeabilized and stained as described above. Images were acquired with a ScanR high-content screening microscope (Olympus). Automated and unbiased image analysis was carried out with the ScanR analysis software (version 2.8.1). Data were exported and processed using Spotfire software (version 10.5.0; Tibco).

## In vitro binding experiments

All in vitro binding experiments were performed in EBC or RIPA buffer as indicated. Recombinant Biotin-tagged K48-linked poly-ubiquitin chains (2–7) and K48-linked poly-ubiquitin chains (2–16) were purchased from R&D Systems and Enzo Life Science, respectively. Recombinant proteins were incubated with Strep-Tactin Sepharose (IBA), Glutathione agarose (Gold Biotechnology) or DYKDDDDK Fab-trap agarose (ChromoTek) overnight on an end-over-end rotator at 4 °C, washed six times with EBC or RIPA buffer, and resuspended in 2×Laemmli sample buffer. For p97, VCF1 or p47 pulldowns, antibodies (0.5 μg/sample) were coupled to agarose beads, incubated with recombinant proteins or cell lysates and washed with the indicated buffer. To assess p97-UFD1-NPL4 binding to ubiquitylated proteins in total cell extracts, U2OS cells were lysed in EBC buffer supplemented with protease and phosphatase inhibitors, sonicated and incubated on ice for 10 min. Lysates were cleared by centrifugation for 10 min at 16,100 × g. Cell lysate (40 μg) was added to recombinant proteins in EBC buffer, incubated overnight on the rotator with DYKDDDDK Fab-trap agarose (ChromoTek), and beads were washed six times.

Peptide pulldowns were performed using Streptavidin High Performance beads (GE Healthcare) and either RIPA or EBC buffer. Peptides containing an N-terminal biotin tag were purchased from GenScript and dissolved according to manufacturer's recommendation. The following peptides were used: VCF1-1 (amino acids (aa) 1–46 of human VCF1): MGGRGADAGSSGGTGPTEGYSPPAASTRAAARAKAR GGGRG; VCF1-2 (aa24-70): AASTRAAARAKARGGGRGGRRNTTPSVPSLR GAAPRSFHPPAAMSER; VCF1-3 (aa51-99): PSLRGAAPRSFHPPAAMS ERLRPRKRRRNGNEEDNHLPPQTKRSSRNPV; VCF1-4 (aa87-128): LPP QTKRSSRNPVFQDSWDTESSGSDSGGSSSSSSSSSINSPD; VCF1-5 (aa116-157): SSSSSSSSINSPDRASGPEGSLSQTMAGSSPNTPQPVPEQSA; VCF1-6 (aa145-186): SLSQTMAGSSPNTPQPVPEQSALCQGLYFHINQTLREAHFH SLQHRGRPLT; VIM (SVIP, aa14-39): PTPDLEEKRAKLAEAAERRQKEAAS; VBM (ATXN3, aa275-295): NLTSEELRKRREAYFEKQQQK.

Peptides were coupled to beads for a minimum of 2 h at 4 °C with rotation, blocked in BSA (New England Biolabs) and washed three times for 1 min, before incubation overnight at 4 °C with whole cell lysate or purified proteins. Beads were then washed three times for 1 min and bound material was eluted in 2×Laemmli sample buffer and analyzed by immunoblotting.

## Affinity purification and mass spectrometry

GFP-Trap agarose (Chromotek) pulldowns were subjected to partial on-bead digestion for peptide elution, by addition of 100 μl of elution buffer (50 mM Tris, pH 7.5; 2 M urea; 2 mM DTT; 20 μg/ml trypsin) and incubation at 37 °C for 30 min. To carry out alkylation, 25 mM CAA was added to the samples, which were then left to digest overnight at room temperature. Digestion was halted by introducing 1% TFA. Purification and desalting of peptides were done using SDB-RPS StageTips. This process required preparing two layers of SDB-RPS with 100 μl of wash buffer (0.2% TFA in H$_2$O), onto which the peptides were loaded. After centrifugation for 5 min at 500 × g, peptides were washed using 150 μl wash buffer, eluted with 50 μl elution buffer (80% ACN; 1% ammonia) and vacuum-dried.

Liquid chromatography-mass spectrometry (LC-MS) analysis was conducted using an EASY-nLC-1200 system (Thermo Fisher Scientific), coupled with a timsTOF Pro spectrometer and a nano-electrospray ion source (Bruker Daltonik). Peptides were loaded onto a 50 cm custom-packed HPLC-column (75 μm inner diameter filled with 1.9 μm Repro-SilPur C18-AQ silica beads, Dr. Maisch). To separate peptides, we followed a linear gradient from 5% to 30% buffer B in 43 min, which then increased to 60% for 7 min, 95% for 5 min, and 5% for a final 5 min at a flow rate of 300 nL/min. Buffer B was a mix of 0.1% formic acid and 80% ACN in LC-MS grade water, while buffer A contained only 0.1% formic acid in LC-MS grade water. The whole gradient cycle lasted for 60 min, and a column oven maintained the column temperature at 60 °C.

Mass spectrometric analysis was carried out in data-dependent (ddaPASEF) mode, as described in ref. [48]. Each cycle incorporated one MS1 survey TIMS-MS and ten PASEF MS/MS scans. Both ion accumulation and ramp time in the dual TIMS analyzer were set to 100 ms. Ion mobility range from 1/K0 = 1.6 Vs/cm$^{-2}$ to 0.6 Vs/cm$^{-2}$ was studied.

Precursor ions for MS/MS analysis were isolated using a 2 Th window for m/z < 700 and 3 Th for m/z > 700, within an m/z range of 100–1.700. Synchronization of the quadrupole switching events was done according to the precursor elution profile of the TIMS device. The collision energy decreased linearly, starting from 59 eV at 1/K0 = 1.6 VS/cm$^{-2}$ to 20 eV at 1/K0 = 0.6 Vs/cm$^{-2}$, corresponding to the rise in mobility. We applied a polygon filter (out of control, Bruker Daltonik) to exclude singly charged precursor ions. Precursors for MS/MS were chosen above an intensity threshold of 1.000 arbitrary units (a.u.) and underwent resequencing until hitting a 'target value' of 20.000 a.u, considering a dynamic exclusion of 40-sec elution.

## Crosslinking mass spectrometry

Purified FLAG-p97 and Strep-HA-VCF1 proteins were mixed in 100 μl buffer (30 mM HEPES, pH 7.5; 300 mM NaCl; 1 mM TCEP), with both proteins at a final concentration of 1 μM. To crosslink the proteins, the MS-cleavable crosslinking agent disuccinimidyl dibutyric urea (DSBU) was added to a final concentration of 3 mM, and incubated at 25 °C for 1 h. The crosslinking reaction was stopped by adding Tris-HCl pH 8.0 to a final concentration of 50 mM and incubating at 25 °C for 30 min. Crosslinked proteins were precipitated with four volumes of ice-cold (−20 °C) acetone, and incubated at −20 °C overnight, after which samples were centrifuged at 15,000 rpm for 5 min. The supernatant was removed and the pellet was dried by exposing the pellet to air at room temperature for a few minutes until it started to appear transparent. Protein pellets obtained from acetone precipitation were dissolved and denatured in 25 μl of denaturation-reduction solution (8 M urea; 5 mM TCEP) at 25 °C for 30 min. Alkylation of cysteine residues was performed by adding 2-chloroacetamide (CAA) to a final concentration of 5 mM and incubating at 25 °C for 20 min, after which 25 μl ABC buffer (20 mM ammonium bicarbonate, pH 8.0) was added to reduce the concentration of urea to 4 M. Lys-C digestion (1:200 w/w) was performed at 25 °C for 3 h, followed by trypsin digestion (1:50 w/w) at 25 °C overnight. Urea was diluted to a final concentration of 0.8 M urea using ABC buffer, after which the samples were further digested with Glu-C (1:200 w/w) at 25 °C overnight.

Peptides were purified using StageTips at high pH as described previously[49]. Quad-layer StageTips were prepared using four punch-outs of C18 material (Sigma-Aldrich, Empore SPE Disks, C18, 47 mm). StageTips were equilibrated using 100 μl of methanol, 100 μl of 80% ACN in 200 mM ammonium hydroxide, and two times 75 μl 50 mM ammonium. Samples were supplemented with one fifth volume of 200 mM ammonium hydroxide (pH > 10), just prior to loading them on StageTips. The StageTips were subsequently washed twice with 150 μl 50 mM ammonium hydroxide, and afterwards eluted using 80 μl of 25% ACN in 50 mM ammonium hydroxide. All samples were dried to completion in protein-LoBind tubes (Eppendorf), using a SpeedVac for 2 h at 60 °C, after which the dried peptides were dissolved using 10 μl of 0.1% formic acid, and stored at −20 °C until MS analysis.

Samples were analyzed on an EASY-nLC 1200 system (Thermo) coupled to an Orbitrap Exploris 480 mass spectrometer (Thermo). Separation of peptides was performed using 20-cm columns (75 μm internal diameter) packed in-house with ReproSil-Pur 120 C18-AQ 1.9 μm beads (Dr. Maisch). Elution of peptides from the column was achieved using a gradient ranging from buffer A (0.1% formic acid) to buffer B (80% acetonitrile in 0.1% formic acid), at a flow of 250 nl/min. The gradient length was 80 min per sample, including ramp-up and wash-out, with an analytical gradient of 60 min ranging from 5% B to 38% B. Analytical columns were heated to 40 °C using a column oven, and ionization was achieved using a NanoSpray Flex NG ion source.

Spray voltage was set to 2 kV, ion transfer tube temperature to 275 °C, and RF funnel level to 50%. Full scan (MS1) range was set to 300–1300 m/z, MS1 resolution to 120,000, MS1 AGC target to "200" (2,000,000 charges), and MS1 maximum injection time to "Auto". Precursors with charges 3–6 were selected for fragmentation using an isolation width of 1.3 m/z, and fragmented using higher-energy collision dissociation (HCD) with normalized collision energy of 27. Precursors were excluded from re-sequencing by setting a dynamic exclusion of 60 s. MS/MS (MS2) AGC target was set to "200" (200,000 charges), MS2 maximum injection time to "Auto", MS2 resolution to 45,000, intensity threshold to 230,000 charges per second, and number of dependent scans to 9.

## Analysis of proteomic data

For analysis of affinity purification mass spectrometry data, raw files obtained in ddaPASEF mode were analyzed with MaxQuant (version 1.6.17.0)[50]. Searches were conducted on the UniProt database (2019 release, UP000005640_9606), implementing a peptide spectral match (PSM) and protein level FDR of 1%. The search criteria required a minimum of seven amino acids, including N-terminal acetylation and methionine oxidation as variable modifications, and cysteine carbamidomethylation as a fixed modification. Enzyme specificity was set to trypsin, permitting a maximum of two missed cleavages. The initial and primary search mass tolerance were set to 70 ppm and 20 ppm, respectively. The identification of peptides was accomplished by aligning four-dimensional isotope patterns between the different runs (MBR), utilizing a 0.7-min window for matching retention time and a 0.05 1/K0 window for ion mobility. For label-free quantification, we used the MaxLFQ algorithm[51] with a minimum ratio count of two. Proteomics data were analyzed using Perseus (version 1.6.15.0)[52], and MaxQuant output tables were cleansed of 'Reverse', 'Only identified by site modification', and 'Potential contaminants' entries before the analysis. Missing data was filled in following a meticulous filtering process, based on a standard distribution (width = 0.3; downshift = 1.8) before performing statistical evaluations. In conducting pairwise comparisons of the proteomics data via a two-sided unpaired $t$ test, we incorporated a 1 or 5% permutation-based FDR to account for multiple hypothesis testing, also including an s0 value[53] of 1 or 2.

For analysis of crosslinking mass spectrometry data, the data were first converted from the Thermo-Finnigan RAW format to Mascot Generic Format (MGF) using pFind Studio version 3.1.0[54]. For this purpose, the RAW files were loaded into pFind, and processed using default data extraction settings and output to MGF enabled. After the pParse and pXtract modules finished the conversion from RAW to MGF, the search in pFind was aborted. To identify DSBU-crosslinked peptides, the MGF files were analyzed using MeroX version 2.0.1.4[55]. The full-length sequences of human p97 (UniProt: P55072) and VCF1 (UniProt: Q969W3) were used for the search. MeroX default settings were used, with exceptions specified below. Digestion specificity was set to C-terminal of arginine with a maximum of 2 missed cleavages, C-terminal of lysine with a maximum of 2 missed cleavages, C-terminal of glutamic acid with a maximum of 2 missed cleavages, and C-terminal of aspartic acid with a maximum of 4 missed cleavages. The global maximum number of missed cleavages increased to 6, to accommodate low off-target activity of Glu-C towards aspartic acid residues. DSBU crosslinking was set to allow KSTY (site 1) to KSTY (site 2) crosslinks. Analysis mode was set to Quadratic (as intended for 1–10 proteins), and minimum score for individual peptides (in quadratic mode) was increased to 5. Processed data was exported to xiViewer format and visualized online using xiView[56].

## Structural modeling of p97 complexes

Structures of complexes between monomeric p97 and human VCF1, SVIP, ATX3, NPL4 and UFD1 were predicted with AlphaFold-Multimer (version 2.3.1)[27,28] based on full-length amino acid sequences from UniProt. In each case, 5–10 models were generated and only the best-ranking model was used for further analysis. In addition, we predicted one model of six human p97 subunits in complex with one copy of human UFD1-NPL4. pLDDT values and PAE plots for these models were visualized with pyMOL (version 1.2r3pre, Schrödinger, LLC) and ChimeraX[57]. The composite model of a p97 hexamer in complex with UFD1, NPL4 and VCF1 was generated by superimposing three copies of the AlphaFold-Multimer prediction of monomeric p97-VRM onto three p97 subunits within the AlphaFold-Multimer model of the hexameric p97-UFD1-NPL4 complex using pyMOL. Buried surface areas were calculated with PDBePISA (http://www.ebi.ac.uk/msd-srv/prot_int/picite.html).

## Surface Plasmon Resonance (SPR)

All SPR experiments were performed at 25 °C on a Biacore T200 instrument equipped with CM5 or CM7 sensor chips (Cytiva, Uppsala, Sweden). SPR running buffers and amine-coupling reagents (N-ethyl-N'-(3-dimethylaminoproply)carbodiimide (EDC), N-hydroxysuccinimide (NHS), and ethanolamine HCl) were purchased from Cytiva. 1x PBS-P (11.9 mM $NaH_2PO_4$-$Na_2HPO_4$, pH 7.4; 137 mM NaCl; 2.7 mM KCl; 0.005% surfactant P20) supplemented with 0.5 mM TCEP was used as running buffer for the immobilization and the binding experiments with Strep-HA-VCF1 full-length proteins. For the VCF1 C-terminal peptide binding studies, 2% DMSO was additionally included in the running buffer. Eight solvent correction samples ranging from 1.5% DMSO to 2.8% DMSO prepared in either PBS-P, 0.5 mM TCEP or Tris-based buffer were used to correct buffer/DMSO mismatches. The CM5 chip was used for the kinetic analysis of the interactions between p97 proteins and VCF1 full-length proteins or VCF1 C-terminal peptides. p97 was immobilized at different densities on the carboxymethyl dextran sensor chip surface by amine-coupling chemistry. For kinetic analysis, p97 was immobilized at low density (~400 RU) on flow cells 2 (Fc2) and 3 (Fc3), respectively, of a CM5 chip. The flow cell 1 (Fc1) remained unmodified and was used as a reference cell for subtraction of systematic instrumental drift. The carboxyl groups were activated for 7 min using 0.1 M NHS and 0.4 M EDC mixed at a 1:1 ratio and a flow rate of 10 µl/min. Short pulses of p97 at 40 µg/ml in 10 mM sodium acetate (pH 5.0) were then flowed over the activated surface until the appropriate immobilization level was reached. The excess activated carboxyl groups were blocked with 1 M ethanolamine, pH 8.5 (7 min injection at a 10 µl/min flow rate). VCF1 C-terminal peptide stocks (150 µM) were prepared in pure DMSO (Sigma Aldrich). Threefold serial dilutions of VCF1 proteins or peptides were prepared in the SPR running buffer from the primary stocks starting from highest concentration of 300 nM (VCF1 C-terminal peptides) or 100 nM (VCF1 full-length proteins). VCF1 peptides and full-length proteins were injected sequentially over all flow cells at a flow rate of 60 µl/min for 120 s. The dissociation rate of the p97-VCF1 complexes was monitored for 400 s. A series of concentrations (2 or 3) were run in duplicates. The experiments were run at least in duplicates to confirm the reproducibility of the assay. Data processing and kinetic fitting were done using the BiaEvaluation software (version 3.2.1, Cytiva, Uppsala, Sweden). The raw sensorgrams were solvent corrected (when appropriate), double referenced (referring to the subtraction of the data over the reference surface and the average of the buffer injections from the binding responses), and the association and dissociation phases over all replicates were globally fitted using a 1:1 interaction model yielding single values for the $k_a$ and the $k_d$. The equilibrium dissociation constant, $K_d$, is the rate of the $k_d$ over the $k_a$. Binding experiments were measured in triplicates. Data processing and fitting was done using the BiaEvaluation software (version 3.2.1, Cytiva, Uppsala, Sweden). Sensorgrams were solvent corrected and double referenced. The equilibrium dissociation contants, $K_d$, were determined

by plotting the equilibrium responses levels (Req) against nucleotide concentrations and fitting to a steady-state model.

## Mass photometry

Mass photometry experiments were performed on a Refeyn One (Refeyn Ltd) mass photometer at room temperature. Measurements were run in triplicates, diluting 3 μl of protein sample (20 nM) in 15 μl PBS buffer loaded on a gasket (Grace Bio-Labs reusable CultureWell gaskets (GBL103250, Merck)) mounted on a glass microscope cover-slip (0107222, Marienfeld). Experiments were performed with $His_6$-p97 alone and in presence of saturating concentrations of full-length Strep-HA-VCF1. The molecular weight was obtained by contrast comparison with known molecular weight mass standard calibrants (NativeMark Unstained Protein Standard (LC0725), Thermo Fischer Scientific) measured on the same buffer and on the same day. Movies were recorded using AcquireMP (Refeyn Ltd, version 2022 R1), with a medium field of view and exposure time of 0.95 ms for the acquisition camera settings. A total of 6000 frames were recorded over a duration of one minute. Movies were processed and analysed with the DiscoverMP software (Refeyn Ltd, version 2022 R1) provided by the instrument manufacturer.

## Analytical gel filtration

For analytical gel filtration, p97 alone (5 μM hexamer), or p97 (5 μM hexamer) with UFD1-NPL4 (5 μM) and VCF1 (5 μM) were mixed in a total volume of 25 μl in SEC buffer (50 mM HEPES, pH 7.4; 150 mM NaCl). Samples were separated by size exclusion chromatography on a Superose 6 3.2/300 column (Cytiva) equilibrated in SEC buffer using an ÄKTA Purifier 10 FPLC system (GE Healthcare). After the column dead volume (0.35 column volumes), fractions were collected (120 μl) and analyzed by immunoblotting.

## ATPase assays

For ATPase assays, 1 ml of 5x assay buffer (250 mM Tris, pH 7.4; 100 mM $MgCl_2$; 5 mM EDTA) was mixed with 3 ml water, 5 μl 0.5 M TCEP and 5 μl 10% Triton X. To 40 ul of this assay buffer, recombinant proteins were added, mixed and incubated for 30 min. Final protein concentrations were as follows: p97 (25 nM), VCF1 (range of 0, 0.025, 1, 10 and 25 nM). The reactions were started by adding 10 μl ATP (final concentration of 375 μM) to the 40 μl reactions in a 96 well plate. After 35 min at room temperature, reactions were stopped by adding 50 μl BIOMOL Green reagent (Enzo Life Sciences). Absorbance at 635 nm was measured after 4 min and the relative ATPase activity calculated. Reactions were performed in duplicates and repeated three times.

## In vitro p97 unfoldase assays

Di-ubiquitin-fused mEos3.2 fluorescent protein (diUb-mEos3.2) and ubiquitylation cascade proteins mUbe1, gpc78-ubc7 fusion and ubiquitin were recombinantly expressed in *E. coli* and purified as described previously[58]. diUb-mEos3.2 substrate was enzymatically polyubiquitylated as described previously[59]. Briefly, His-diUb-mEos3.2 substrate was incubated with mUbe1 (2 μM), gp78-ubc7 (20 μM), ubiquitin (400 μM) and ATP (10 mM) overnight at 37 °C before the polyubiquitylated substrate ($diUb^n$-mEos3.2) was purified from free ubiquitin chains by Ni-NTA affinity chromatography (HisTrap FF, 1 mL; Cytiva) and size exclusion chromatography (Superdex 10/300 200 GL; Cytiva). mEos3.2 was partially photo-converted from green to red light-emitting forms by irradiation with a UV-lamp (365 nm; Blak-Ray, B-100AP) for 2 h at 4 °C. The unfolding activity of p97 was monitored in vitro by measuring the decrease in $diUb^n$-mEos3.2 fluorescence. For this, $diUb^n$-mEos3.2 (35 nM), p97 (100 nM hexamer), UFD1-NPL4 (500 nM) and VCF1 (300 nM) were mixed as indicated in Eos assay buffer (50 mM HEPES, pH 7.5; 150 mM KCl; 2 mM $MgCl_2$; 5% glycerol; 1 mM DTT) and pre-warmed to 37 °C for 5 min before ATP (2 mM) was added to start the reaction. Unfolding of red mEos3.2 fluorescence was measured every 15 s in a Varian Cary Eclipse spectrofluorometer (excitation: 500 nm; emission: 520 nm) and normalized to the first value after addition of ATP.

## Reporting summary

Further information on research design is available in the Nature Portfolio Reporting Summary linked to this article.

## Data availability

The mass spectrometry proteomics data generated in this study have been deposited to the ProteomeXchange Consortium[60] via the Proteomics Identifications (PRIDE) partner repository (http://www.ebi.ac.uk/pride) under dataset ID PXD043565 (Supplementary Data 1–3) and dataset ID PXD043563 (Fig. 3e). The AlphaFold models, results and input files generated in this study have been deposited to the Electronic Research Data Archive at University of Copenhagen (https://sid.erda.dk/share_redirect/h6SgLgmOGi). All other data supporting the findings of this study are available within the article and supplementary information. Source data are provided with this paper.

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

## Acknowledgements

We thank Olaf Stemmann and Yves Durocher for providing reagents, Satya Pentakota and Dandan Xue for technical support with protein production and AlphaFold applications, respectively, and members of the Mailand laboratory for helpful discussions. This work was supported by grants from Novo Nordisk Foundation (grants no. NNF14CC0001 (M.L.N., J.P.D., G.M. and N.M.) and NNF18OC0030752 (N.M)), European Union's Horizon 2020 research and innovation program (Marie-Skłodowska-Curie grant agreement no. 812829 (aDDRess) (A.S.M.) and European Research Council (ERC) grant agreement no. 715975 (J.P.D)), Lundbeck Foundation (grant no. R223-2016-281 (N.M.) and R303-2018-3212 (S.H.)), Danish Cancer Society (N.M.), Independent Research Fund Denmark (grant no. 0134-00048B (N.M.)), and Danish National Research Foundation (grant no. DNRF-115 (N.M.)), German Research Foundation (DFG; SFB1430; Project-ID 424228829 (H.M.)) and EMBO (ALTF 1149-2020 (B.B.)).

## Author contributions

Conceptualization: A.S.M. and N.M.; Methodology: A.S.M., S.H., M.W., A.M., B.L.M., I.A.H., G.M. and N.M.; Investigation: A.S.M., S.H., M.W., A.M., B.L.M., D.T., J.v.d.B., B.B. and I.A.H.; Writing – Original Draft: N.M.; Writing – Review and Editing: All authors; Supervision: M.L.N., J.P.D., H.M., G.M. and N.M.; Project Administration: N.M.; Funding Acquisition: S.H. and N.M.

## Competing interests

The authors declare no competing interests.
