## [Peer Review File · Nature Communications]

VCF1 is an unconventional p97/VCP cofactor promoting recognition of ubiquitylated p97-UFD1-NPL4 substratesREVIEWER COMMENTS

Reviewer #1 (Remarks to the Author):

In this study, Mirsanaye identified FAM104A as an unconventional co-factor of p97 that stimulates ubiquitin-dependent degradation of p97 by regulating its interaction with the ubiquitin-selective adaptor UFD1/NPL4. The authors provide compelling biochemical and structural data that support a regulatory function of FAM104A in the context of p97, which they therefore renamed to VCF1 (for VCP/p97 Cofactor FAM104 1). They found that the C-terminal helix in VCF1 is responsible for binding to N-domain of p97. At a molecular level it is not clear how VCF1 promotes interaction of p97 with UFD1/NPL4. This is apparently not trivial as it also competes (at supraphysiological levels) with UFD1/NPL4 for binding. The authors speculate about possible molecular mechanisms, but I think it is fair to say that this is far from clear. This is an interesting study that is well documented with high quality data.

Main comments

In Fig 1A the authors show that VCF1 is a nuclear protein. This is based on the localization of a GFP-VCF1 fusion protein. It is feasible that the GFP fusion (due to its size or due to sterically hinderance) influences the localization of VCF1. According to the Protein Atlas the localization of VCF1 is nuclear and cytosolic. Although I would put higher value on the observations presented in this paper than the annotation in the Protein Atlas, I do think it would be important to confirm the localization for endogenous VCF1. Also, do the authors know if VCF1 has a nuclear localization signal? Considering its small size and disordered structure, I would otherwise expect it to freely diffuse between the cytosol and nucleus.

If the authors can confirm that the endogenous VCF1 resides primarily in the nucleoplasm, this raises the question if its effect on proteasomal degradation is also confined to the nuclear compartment. This could be addressed by analyzing the effect on compartment specific GFP-based proteasome substrates that are available. If the effect is indeed specific for the nuclear compartment, implications should be discussed. One would for example in that case expect that VCF1 is not involved in ERAD, which is an important function of p97.

Fig 1G. The authors conclude that VCF1 binds to the N domain of p97. However, the pulldown seems to be much more efficient for the N-D domain of p97 as compared to the N domain on its own. Please comment on this observation. Does the structural analysis provide any explanation?

In Fig 5A and B is shown that depletion of NPL4 + VCF1 results in a much stronger effect than depletion of NPL4 or VCF1. A straightforward interpretation would be that VCF1 and NPL4 act independently of

each other and that therefore combined depletion results in an additive effect. If they were acting in concert, one would expect that the combined depletion gave the same effect as the strongest effect by depleting VCF1 or NPL4. The authors argue differently and, even though I can follow their reasoning and do not disagree, I think their interpretation is somewhat counterintuitive and may confuse readers. It would be helpful if they could explain this a little bit better.

Depletion and overexpression of VCF1 both result in inhibition of degradation. The authors propose (if I understood this correctly) that VCF1 may prevent binding of other co-factors to p97 and thereby promoting p97 complexes that only contain the UFD1/NPL4 co-factor and are therefore solely dedicated to ubiquitin-dependent degradation. However, they argue that at higher supraphysiological concentrations it may also start to compete with UFD1/NPL4. If this were the mechanism, would one not predict that depletion of VCF1 in NPL4-depleted cells would partially restore p97 function instead of worsening it as the ratio between VCF1 and NPL4 would be restored?

Have the authors analyzed or is it known from the literature whether VCF1 is regulated by proteotoxic or genotoxic stress conditions? Given the important role of p97 in proteostasis and DNA repair, it would be interesting to know if regulation of VCF1 is used to stimulate p97-dependent degradation under specific conditions.

A lot of the seminal work on p97/Cdc48 has been done in yeast. P97, UFD1, NPL4 are all conserved in yeast. Have the authors any information on a yeast ortholog of VCF1? According to the databases VCF1 is only present in chordates. What are the evolutionary implications for the function of such a conserved and critical complex as p97?

This may be a matter of taste, but I really don't see the point in confusing the literature and various databases by given FAM104A a new name. I don't think that a new function for a known protein should automatically result in renaming it unless the original name can lead to misunderstandings. For me there is no need to have a new synonym for FAM104A.

Reviewer #2 (Remarks to the Author):

The manuscript by Mirsanaye* and Hoffman* et al. reports the discovery and biochemical characterization of the novel p97 accessory cofactor, VCF1. p97 is an abundant and essential enzyme that unfolds ubiquitylated substrates prior to their degradation. p97 interacts with many accessory cofactors to mediate this process, the best characterized of which is the Ufd1-Npl4 heterodimer (UN). Here, the authors identified VCF1 as a novel binding partner to p97-UN complexes and present

biochemical experiments that indicate its role in enhancing p97-UN interactions with ubiquitylated substrates.

The manuscript is well written and the results are interesting, but the quality of the manuscript is diminished by the relative lack of biological insights and apparent contradictory data for VCF1. In order to improve the manuscript to a more suitable state for publication, I encourage the authors to consider the following points:

1. Biological insight is limited to observations that the model ubiquitylated substrate, Ub(G76V)-GFP, significantly accumulates upon co-depletion of both VCF1 and NPL4 (Fig. 5A). While this result is convincing in terms of a role of VCF1 in enhancing Ub-substrate turnover, there are no indications of a meaningful biological significance for this role.

The manuscript suggests a relationship between increased Ub binding in the presence of VCF1 and increased Ub-substrate turnover in cells, but more direct evidence is needed. To this end, it would be ideal to demonstrate whether VCF1 enhances p97-mediated unfolding using *in vitro* assays.

2. There are many inconsistencies and unexplained results with IP and immunoblotting experiments in various figures as follows:

a. It is unclear why a band for p97 is not observed in the second input lane of Fig. 1E.

b. The authors conclude that "UFD1-NPL4 co-purified with immobilized VCF1 in a manner that was fully dependent on the presence of p97", but Fig. 4B (3rd lane) shows that VCF1 co-purifies with Ufd1 in the absence of p97.

c. The authors conclude that VCF1 interacts more tightly with the p97 N-domain than other reported p97 cofactors, including p47, and show that moderate concentrations of VCF1 were clearly permissive for the simultaneous binding of VCF1 and UFD1-NPL4, p47, or FAF1 to p97. However, in Fig. S4B (4rd lane), it is unclear why VCF1 did not co-purify with p47 in the presence of p97.

d. The authors used SPR and pull-down experiments to verify that the N167A mutation in both full-length VCF1 and the VCF1-6 peptide abrogated binding to p97 (Fig. 2F; Fig. S2D, E) and concluded that UN co-purified with immobilized VCF1 in a manner that was fully dependent on the presence of p97. However, Fig. 5C (4rd lane) shows that VCF1 N167A was pulled down by p97, which is contradictory to their conclusions.

Minor comments:

1. Can the authors comment on the potential significance of C1QBP and HIST1H3A associating with VCF1-GFP co-IP? (Fig. 1C and S1C)

2. The crosslinking mass spectrometry results suggest that VCF1 interacts with all domains of p97 (Fig. 3E). Is there support for this through AlphaFold-Multimer predictions?

3. "2,5 mM Desthiobiotin" should be corrected to "2.5 mM Desthiobiotin." (line 487)

Reviewer #3 (Remarks to the Author):

I carefully read the manuscript by Mirsanaye et al. on the regulation of human p97 by cofactors. It is a really interesting manuscript, well written and understandable in every part. P97 is an important player in the proteasome degradation pathway. The authors discover and characterize a novel cofactor, VCF1, which has an unconventional interaction mode and which strongly regulates the activity of p97 in the UFD1-NPL4 pathway. The authors lucidly investigate the interactions between p97 and VCF1, as well the functional modulation, using a battery of methodologies. I fully endorse the publication, and I have a few minor comments and suggestions:

1) The models of VCF1-p97 generated with AF2 are somewhat validated by crosslinking MS, alanine scanning mutagenesis, domain deletion, sequence conservation, and other mutational assays. These models overall are making a lot of sense and are very useful. The authors report that the regions outside the C-terminal part of VCF1 have a very low pLDDT score, and this is in agreement with the intrinsically disordered character of the protein. Of course the lack of confidence does not guarantee for lack of structure, and in fact cross-links are also present between the N-terminal and p97. It could be interesting to see if AF can predict any structure modeling the central peptide of VCF1 (which heavily crosslinks with the rest of p97) and p97, although I suspect that these are transient interactions, as reported by the authors.

2) The AlphaFold results and methods are a bit slim, and it is important to report a bit more details: it would be interesting to see the Sequence coverage, the plot of the pLDDT, and especially the PAE plot for the VCF1-p97 and the full complex, to have an idea of the quality of the assembly performed by AF multimer. How many models are generated? Is there any remarkable difference between them? Also, all models and results, as well as the input files, from AF should be made available, for instance by depositing in Zenodo.

3) Can the authors report and comment on the intra-molecular crosslinks, ie, those occurring within p97 and within VCF1? What is the accuracy of crosslinks if they are mapped on p97 hexameric structure? They should provide the cross-link database available to the reader.

4) In Suppl fig #, paned A, the model confidence scale reads "Very low (pLDDT>50)" instead it should be "Very low (pLDDT<50)"

Point-by-point reply to the referees' comments

We thank the reviewers for their enthusiastic and constructive feedback on our study. In the revised version of our manuscript, we have included the results of new experiments performed on the basis of the reviewers' helpful suggestions, and several points have been clarified in the text and figures. Collectively, we believe the new additions and changes to the manuscript address the referees' concerns. In the following, we provide a detailed point-by-point response to the referee reports. The reviewers' comments (reproduced in their entirety) are shown in bold and italicized text, while our responses are in plain text. With the new additions, we hope the reviewers find our paper improved and ready for publication in Nature Communications.

Reviewer #1:

In this study, Mirsanaye identified FAM104A as an unconventional co-factor of p97 that stimulates ubiquitin-dependent degradation of p97 by regulating its interaction with the ubiquitin-selective adaptor UFD1/NPL4. The authors provide compelling biochemical and structural data that support a regulatory function of FAM104A in the context of p97, which they therefore re coined to VCF1 (for VCP/p97 Cofactor FAM104 1). They found that the C-terminal helix in VCF1 is responsible for binding to N-domain of p97. At a molecular level it is not clear how VCF1 promotes interaction of p97 with UFD1/NPL4. This is apparently not trivial as it also competes (at supraphysiological levels) with UFD1/NPL4 for binding. The authors speculate about possible molecular mechanisms, but I think it is fair to say that this is far from clear. This is an interesting study that is well documented with high quality data.

Main comments

In Fig 1A the authors show that VCF1 is a nuclear protein. This is based on the localization of a GFP-VCF1 fusion protein. It is feasible that the GFP fusion (due to its size or due to sterically hinderance) influences the localization of VCF1. According to the Protein Atlas the localization of VCF1 is nuclear and cytosolic. Although I would put higher value on the observations presented in this paper than the annotation in the Protein Atlas, I do think it would be important to confirm the localization for endogenous VCF1. Also, do the authors know if VCF1 has a nuclear localization signal? Considering it small size and disordered structure, I would otherwise expect it to freely diffuse between the cytosol and nucleus.

Unfortunately, our VCF1 antibody is not suitable for detection of endogenous VCF1 in immunofluorescence and subcellular fractionation experiments, possibly due to its low expression level. However, we found that VCF1 fused to a small tag (Strep-HA) also localizes almost exclusively to the nucleus (new Figure S1c), alleviating the valid concern that the large size of the GFP tag may interfere with the proper subcellular localization of VCF1. Moreover, as described in our response to the next point below, we obtained new data suggesting that the function of endogenous VCF1 in promoting p97 function is mainly confined to the nuclear compartment (new Figure S5d), in line with VCF1 being predominantly localized in the nucleus. We agree with the reviewer that the VCF1 subcellular localization data in the Human Protein Atlas (HPA) should be interpreted with caution. Indeed, the antibody HPA used for immunofluorescence analysis of VCF1 recognizes several

bands in immunoblots, none of which correspond to the size of VCF1 (<https://www.proteinatlas.org/ENSG00000133193-FAM104A/summary/antibody#WB>), so the validity of the HPA subcellular localization data for VCF1, which additionally lack knockdown/knockout controls, is somewhat uncertain.

Using truncated forms of VCF1, we found that its N-terminal portion harbors the sequence determinant(s) underlying VCF1 nuclear localization (new Figure S1c). However, while VCF1 contains a conserved bipartite nuclear localization signal (NLS)-like sequence in this region (amino acids (aa) 74-93; http://slim.icr.ac.uk/proviz/proviz.php?uniprot_acc=Q969W3), inactivation of this potential NLS by in-frame deletion of the stretch of basic residues in aa74-79 only led to a modest appearance of GFP-VCF1 in the cytoplasm unlike deletion of larger portions of the VCF1 N-terminal region (Reviewer Figure 1; compare with new Figure S1c), suggesting that additional N-terminal sequences also contribute to targeting VCF1 to the nucleus.

Reviewer Figure 1. Inactivation of a predicted bipartite NLS in the VCF1 N-terminus has little impact on its nuclear localization

Representative images of U2OS cells transfected with expression construct encoding GFP-VCF1(Δ 74-79). Scale bar, 10 μ M.

If the authors can confirm that the endogenous VCF1 resides primarily in the nucleoplasm, this raises the question if its effect on proteasomal degradation is also confined to the nuclear compartment. This could be addressed by analyzing the effect on compartment specific GFP-based proteasome substrates that are available. If the effect is indeed specific for the nuclear compartment, implications should be discussed. One would for example in that case expect that VCF1 is not involved in ERAD, which is an important function of p97.

This is a good point. We tested this by generating and using a cell line stably expressing the Ub(G76V)-GFP reporter fused to a nuclear export sequence (NES), rendering the resulting Ub(G76V)-GFP-NES protein largely cytoplasmic (new Figure S5d). Interestingly, unlike the parental Ub(G76V)-GFP protein that is mainly upregulated in the nucleus when VCF1 is depleted (Figure 5a), Ub(G76V)-GFP-NES showed no discernible accumulation upon VCF1 knockdown, whereas p97 inhibition led to strong Ub(G76V)-GFP-NES accumulation in the cytoplasm as expected (new Figure S5d). This suggests that the role of VCF1 in promoting turnover of p97 client proteins is mainly confined to the nucleus, in line with the predominantly nuclear localization of ectopic VCF1 (Figure 1a; new Figure S1c).

Fig 1G. The authors conclude that VCF1 binds to the N domain of p97. However, the pulldown seems to be much more efficient for the N-D domain of p97 as compared to the N

domain on its own. Please comment on this observation. Does the structural analysis provide any explanation?

Unlike full-length p97 and the p97 N-D1 fragment, the isolated p97 N domain does not engage in hexamer formation, which is mediated by the D1 domain ¹. Accordingly, while VCF1 interacts with hexamers of p97 WT and p97 N-D1, it only pulls down p97 N domain monomers, thus explaining why considerably less p97 N protein co-precipitates with Strep-HA-VCF1 relative to p97 N-D1 (Figure 1g). We now clarify this in the legend for Figure 1g (page 42).

In Fig 5A and B is shown that depletion of NPL4 + VCF1 results in a much stronger effect than depletion of NPL4 or VCF1. A straightforward interpretation would be that VCF1 and NPL4 act independently of each other and that therefore combined depletion results in an additive effect. If they were acting in concert, one would expect that the combined depletion gave the same effect as the strongest effect by depleting VCF1 or NPL4. The authors argue differently and, even though I can follow their reasoning and do not disagree, I think their interpretation is somewhat counterintuitive and may confuse readers. It would be helpful if they could explain this a little bit better.

The interpretation of these data is complicated somewhat by the fact that we are not able to quantitatively deplete NPL4 (Figure S5b), likely because it is essential ². This gives rise to a situation where NPL4 knockdown only moderately impairs Ub(G76V)-GFP degradation (Figure 5a,b), despite the essential role of p97-UFD1-NPL4 in this process. We agree with the reviewer that the additive impact of VCF1 knockdown and partial NPL4 depletion in stabilizing Ub(G76V)-GFP could in principle reflect that VCF1 and UFD1-NPL4 act in independent pathways of p97-dependent protein degradation, as a straightforward interpretation of the data in Figure 5a,b. However, we consider this unlikely since the VCF1-p97 complex on its own does not associate with and unfold ubiquitylated proteins (Figure 5d,e; new Figure S6b; Figure S6d), and degradation of the Ub(G76V)-GFP model substrate is not known to involve primary p97 cofactors other than UFD1-NPL4. Our data show that VCF1 stimulates p97-UFD1-NPL4 interaction with poly-ubiquitin chains and becomes rate-limiting for degradation of the p97-UFD1-NPL4 substrate Ub(G76V)-GFP when NPL4 levels are reduced (Figure 5c-e; Figure S5e). The additive effect of VCF1 and NPL4 depletion in stabilizing Ub(G76V)-GFP is consistent with these findings and our model that VCF1 functions as a p97 cofactor that stimulates p97-UFD1-NPL4-dependent turnover of ubiquitylated proteins. We have tried to better explain this rationale in the revised manuscript. However, as we cannot formally rule out that VCF1 and UFD1-NPL4 might act in separate branches of p97-dependent degradation of ubiquitylated proteins, we now also mention this possibility in the discussion, in keeping with the reviewer's point.

Depletion and overexpression of VCF1 both result in inhibition of degradation. The authors propose (if I understood this correctly) that VCF1 may prevent binding of other co-factors to p97 and thereby promoting p97 complexes that only contain the UFD1/NPL4 co-factor and are therefore solely dedicated to ubiquitin-dependent degradation. However, they argue that at higher supraphysiological concentrations it may also start to compete with UFD1/NPL4. If this were the mechanism, would one not predict that depletion of VCF1 in NPL4-depleted cells would partially restore p97 function instead of worsening it as the ratio between VCF1 and NPL4 would be restored?

As described above, we have shown that VCF1 stimulates poly-ubiquitin binding by the p97-UFD1-NPL4 complex, and we believe this is a primary mechanism by which VCF1 supports p97 functions. The possibility that VCF1 may additionally shield the p97-UFD1-NPL4 complex from binding by other cofactors, which we mentioned in the discussion section, remains speculative at this point and not directly supported by data. Thus, to avoid confusion, we have removed this statement from the revised manuscript.

As shown in Figure S4i, VCF1 is able to partially outcompete NPL4 binding to p97 *in vitro*, but only when present at high concentrations exceeding that of NPL4. However, as we point out in the manuscript this clearly represents a non-physiological situation, as the stoichiometry between endogenous NPL4 and VCF1 is around 100:1 in HeLa cells³. Thus, at endogenous expression levels VCF1 is, in all likelihood, unable to compete with UFD1-NPL4 for binding to p97. Indeed, we observe no appreciable change in the interaction between endogenous NPL4 and p97 upon VCF1 depletion (data not shown). Using NPL4 siRNA, we are only able to achieve ~50% reduction in NPL4 abundance (Figure S5b), thus even under these conditions NPL4 would be predicted to be present at a substantially higher copy number than VCF1. Accordingly, we do not expect that depleting VCF1 under these conditions restores p97 functionality by increasing its binding to UFD1-NPL4, in line with the observation that VCF1 knockdown aggravates the Ub(G76V)-GFP degradation defect in NPL4-depleted cells rather than alleviating it (Figure 5a,b).

Have the authors analyzed or is it known from the literature whether VCF1 is regulated by proteotoxic or genotoxic stress conditions? Given the important role of p97 in proteostasis and DNA repair, it would be interesting to know if regulation of VCF1 is used to stimulate p97-dependent degradation under specific conditions.

This is an interesting suggestion that we considered. To this end, we monitored the impact of a wide panel of proteotoxic and genotoxic agents on VCF1 expression and subcellular localization. We found that none of these treatments had any overt impact on the expression level of endogenous VCF1 and subcellular localization of stably expressed GFP-VCF1 (an example is shown in Reviewer Figure 2A,B below). Likewise, we are not aware of data in the literature indicating that VCF1 is regulated upon stress conditions. Thus, at this point we do not have concrete evidence to suggest that VCF1 status is regulated in a stress-dependent manner, although this remains a distinct possibility.

Reviewer Figure 2. Impact of cellular stresses on VCF1 expression and subcellular localization
A. Immunoblot analysis of U2OS cells collected 1 h after treatment with indicated stress-inducing agents. 5-AzadC, 5-Aza-2'-deoxycytidine, a DNA-protein crosslink-inducing agent. **B.** Representative images of U2OS/GFP-VCF1 WT cells treated with Doxycycline (DOX) for 24 h and fixed 1 h after the indicated treatments. Scale bars, 10 μ M.

A lot of the seminal work on p97/Cdc48 has been done in yeast. P97, UFD1, NPL4 are all conserved in yeast. Have the authors any information on a yeast ortholog of VCF1? According to the databases VCF1 is only present in chordates. What are the evolutionary implications for the function of such a conserved and critical complex as p97?

While VCF1-like proteins are present in both vertebrates and many invertebrates (incl. *Drosophila* and other insects), recognizable VCF1 orthologues seem to be absent in yeast. In general, the p97 system in higher eukaryotes is considerably more complex than the corresponding Cdc48 system in yeast, and during evolution some of the regulatory layers migrated from Cdc48 and UFD1-NPL4 to additional cofactors to enable more plasticity in the system. Importantly, yeast Cdc48-Ufd1-Npl4 is much more proficient towards poly-ubiquitylated substrates than mammalian p97-UFD1-NPL4⁴, thus additional cofactors are needed in mammals to support optimal p97-UFD1-NPL4 activity. A prime example in this context is the p97 cofactor FAF1, which like VCF1 has been shown to enhance p97-UFD1-NPL4 interactions with ubiquitin, albeit by a different mechanism⁴, and which is also not present in yeast. Thus, there is a precedent for important accessory p97 cofactors such as FAF1, which like VCF1 enhance p97-UFD1-NPL4's affinity for ubiquitylated substrates in mammalian cells but are not present in yeast.

This may be a matter of taste, but I really don't see the point in confusing the literature and various databases by given FAM104A a new name. I don't think that a new function for a known protein should automatically result in renaming it unless the original name can lead to misunderstandings. For me there is no need to have a new synonym for FAM104A.

We fully respect the reviewer's opinion but would like to point out that naming protein-coding genes based on a key normal function of the gene product, as overseen by the Human Genome Organization (HUGO), is standard procedure (for more information, please see <https://www.genenames.org/about/guidelines/>). At the same time, maintaining generic 'placeholder' protein names such as FAM104A is strongly discouraged by HUGO. Indeed, shortly after our manuscript was uploaded to bioRxiv, we were contacted by HUGO with the aim to update the nomenclature for FAM104A, in light of our discovery that it defines a novel p97 cofactor. VCF1 has now been adopted as the official name for FAM104A (https://www.genenames.org/data/gene-symbol-report/#!/hgnc_id/HGNC:25918), thus at this point it would not be helpful to revert to the FAM104A nomenclature.

Reviewer #2:

The manuscript by Mirsanaye* and Hoffman* et al. reports the discovery and biochemical characterization of the novel p97 accessory cofactor, VCF1. p97 is an abundant and essential enzyme that unfolds ubiquitylated substrates prior to their degradation. p97 interacts with many accessory cofactors to mediate this process, the best characterized of which is the Ufd1-Npl4 heterodimer (UN). Here, the authors identified VCF1 as a novel binding partner to p97-UN complexes and present biochemical experiments that indicate its role in enhancing p97-UN interactions with ubiquitylated substrates.

The manuscript is well written and the results are interesting, but the quality of the manuscript is diminished by the relative lack of biological insights and apparent

contradictory data for VCF1. In order to improve the manuscript to a more suitable state for publication, I encourage the authors to consider the following points:

1. Biological insight is limited to observations that the model ubiquitylated substrate, Ub(G76V)-GFP, significantly accumulates upon co-depletion of both VCF1 and NPL4 (Fig. 5A). While this result is convincing in terms of a role of VCF1 in enhancing Ub-substrate turnover, there are no indications of a meaningful biological significance for this role.

The manuscript suggests a relationship between increased Ub binding in the presence of VCF1 and increased Ub-substrate turnover in cells, but more direct evidence is needed. To this end, it would be ideal to demonstrate whether VCF1 enhances p97-mediated unfolding using *in vitro* assays.

The Ub(G76V)-GFP substrate is a widely used and well characterized model that faithfully reflects endogenous p97 substrates without the need to induce stress conditions that can affect the analysis. Despite careful studies of the impact of depleting VCF1 on a range of known nuclear p97-driven responses, we have not been able to pinpoint a specific p97-dependent cellular process whose integrity critically relies on VCF1 function. This is likely due to considerable in-built redundancy in the p97 system and partially overlapping functions of accessory cofactors in facilitating p97-UFD1-NPL4-dependent extraction and turnover of a wide range of ubiquitylated targets. Indeed, the ability of VCF1 to stimulate p97-UFD1-NPL4 interaction with ubiquitin conjugates is functionally similar to that of other accessory cofactors for the p97-UFD1-NPL4 complex, including FAF1 and UBXN7.

As suggested by the reviewer, we performed *in vitro* assays⁵ to analyze the effect of VCF1 on p97 unfoldase activity. We found that the VCF1-p97 complex on its own does not unfold a poly-ubiquitylated fluorescent (mEos3.2) p97 substrate (new Figure S6b), consistent with the lack of ubiquitin binding by VCF1-p97 (Figure 5d,e; Figure S6d). In these assays, the addition of VCF1 did not potentiate the already very efficient unfoldase activity of p97-UFD1-NPL4 towards poly-ubiquitylated mEos3.2 (new Figure S6b), despite VCF1 stimulates the affinity of p97-UFD1-NPL4 for poly-ubiquitin chains (Figure 5c-e). This likely reflects a limitation of the *in vitro* assay including the required use of a highly ubiquitylated recombinant mEos3.2 protein that represents an optimal substrate for p97-UFD1-NPL4-mediated unfolding, thus possibly masking roles of accessory cofactors such as VCF1 in facilitating this process. These data further suggest that VCF1 does not stimulate p97 unfoldase activity *per se* but may be important for promoting the processing of p97 client proteins under more restrictive physiological conditions in the cell, e.g. when ubiquitin chain lengths or linkage landscapes on cellular p97 substrates are sub-optimal for efficient p97-UFD1-NPL4 recruitment⁶⁻⁸ or when p97-UFD1-NPL4 availability is reduced (Figure 5a,b; Figure S5b). Our discovery of VCF1 as an important novel component of the ubiquitin-proteasome system provides a solid foundation to further explore its precise physiological relevance in promoting and regulating the highly complex p97 system.

2. There are many inconsistencies and unexplained results with IP and immunoblotting experiments in various figures as follows:

a. It is unclear why a band for p97 is not observed in the second input lane of Fig. 1E.

This panel was unfortunately mislabeled, as no p97 protein was added in lane 2. We apologize for this error, which has now been corrected, and we thank the reviewer for bringing this to our attention.

b. The authors conclude that “UFD1-NPL4 co-purified with immobilized VCF1 in a manner that was fully dependent on the presence of p97”, but Fig. 4B (3rd lane) shows that VCF1 co-purifies with Ufd1 in the absence of p97.

While the level of co-purified VCF1 is only slightly higher in lane 3 than in the negative control (lane 1) in Figure 4b, we sometimes observe modest p97-independent enrichment of purified VCF1 in UFD1 pulldowns *in vitro*, as in Figure 4b. Because the Strep-HA-VCF1 protein used in this experiment was purified from human HEK293T cells, we suspect that its apparent weak p97-independent interaction with UFD1 in these *in vitro* binding assays with recombinant proteins might simply be due to a low level of p97 co-purifying with Strep-HA-VCF1, considering also that no enrichment of UFD1 was observed for the VCF1 N167A mutant (lane 4 in Figure 4b). The statement highlighted by the referee refers to Figure 4c, which we believe does show clearly that enrichment of UFD1 (and NPL4) in VCF1 pulldowns is virtually entirely dependent on p97 and the p97-VCF1 interaction. Still, we moderated the wording of the text highlighted by the referee, so that it now reads as follows: “UFD1-NPL4 co-purified with immobilized VCF1 in a manner that was dependent on the presence of p97” (page 10).

c. The authors conclude that VCF1 interacts more tightly with the p97 N-domain than other reported p97 cofactors, including p47, and show that moderate concentrations of VCF1 were clearly permissive for the simultaneous binding of VCF1 and UFD1-NPL4, p47, or FAF1 to p97. However, in Fig. S4B (4rd lane), it is unclear why VCF1 did not co-purify with p47 in the presence of p97.

Three copies of p47 can bind each p97 hexamer via a bipartite interaction involving the SHP box and UBX domain in p47^{9,10}, and such a complex is likely non-permissive for the simultaneous binding of VCF1. At low concentrations (as in lane 4 in Figure S4b; VCF1:p47 ratio approx. 1:3), VCF1 may not be able to outcompete the tight bipartite p47 binding to the p97 N domain, and we have no evidence to suggest that VCF1 interacts directly with p47; accordingly, little if any VCF1 is detectable in the p47 IP in lane 4 in Figure S4b. At somewhat higher levels (lane 5 in Figure S4b; VCF1:p47 ratio approx. 2:1), VCF1 starts to compete with p47 for binding to p97, hence VCF1 now becomes detectable in p47 IPs from binding reactions containing p47, p97 and VCF1. At high concentrations (lane 6 in Figure S4b; VCF1:p47 ratio approx. 8:1), VCF1 more efficiently outcompetes p47 binding to p97, hence less p97 (and VCF1) co-precipitates with p47 under these conditions.

d. The authors used SPR and pull-down experiments to verify that the N167A mutation in both full-length VCF1 and the VCF1-6 peptide abrogated binding to p97 (Fig. 2F; Fig. S2D, E) and concluded that UN co-purified with immobilized VCF1 in a manner that was fully dependent on the presence of p97. However, Fig. 5C (4rd lane) shows that VCF1 N167A was pulled down by p97, which is contradictory to their conclusions.

Although the N167A point mutation strongly decreases the affinity of VCF1 for p97 (Figure 2f; Figure S2d,e), we occasionally observe slight residual binding of the VCF1 N167A mutant to p97, particularly in our *in vitro* binding assays using high amounts of both proteins and mild buffer conditions (e.g. Figure 5c). However, even under these conditions it is clear

that the N167A mutant is strongly impaired for binding to p97. The Strep-HA-VCF1 blot of the FLAG IP originally shown in Figure 5c was somewhat overexposed, and we have therefore exchanged it with a lower exposure that we believe more faithfully shows the strong reduction in VCF1 binding to p97 caused by the N167A mutation. Importantly, even if the N167A may retain very weak residual binding to p97, we found that it is functionally fully deficient for stimulating p97-UFD1-NPL4 binding to poly-ubiquitin chains (Figure 5c).

Minor comments:

1. Can the authors comment on the potential significance of C1QBP and HIST1H3A associating with VCF1-GFP co-IP? (Fig. 1C and S1C)

Unlike p97, C1QBP and histone H3 were not consistently detected as significant hits across all our VCF1 interactome analyses (Figure 1c; Figure S1d; Figure 2e; Supplementary Data 1-3). We performed co-IP experiments to further evaluate whether C1QBP and histone H3 interact with VCF1. While we were unable to detect C1QBP in VCF1 IPs, we observed a weak interaction between VCF1 and histone H3 (Reviewer Figure 3A). This could suggest that VCF1 interacts with chromatin and may be involved in promoting p97 functions in the context of chromatin. Indeed, a weak but detectable pre-extraction-resistant nuclear GFP-VCF1 signal was evident in microscopy (Reviewer Figure 3B), consistent with VCF1 associating with chromatin. However, future studies will be needed to explore whether and how VCF1 is involved in regulating p97 functions in chromatin.

Reviewer Figure 3. VCF1 interaction with Histone H3 and chromatin

A. GFP IPs from U2OS/GFP-VCF1 WT cells treated or not with Doxycycline (DOX) for 16 h were immunoblotted with C1QBP, Histone H3 and GFP antibodies. **B.** Representative images of U2OS/GFP-VCF1 WT cells treated with Doxycycline (DOX) for 24 h and subjected or not to pre-extraction with 0.5% Triton X-100. Scale bars, 10 μM.

2. The crosslinking mass spectrometry results suggest that VCF1 interacts with all domains of p97 (Fig. 3E). Is there support for this through AlphaFold-Multimer predictions?

As shown in Figure S3a, our AlphaFold-Multimer predictions indeed support that in addition to the high affinity interaction between the VCF1 VRM and the p97 N domain the unstructured regions of VCF1 may engage in interactions with the D1 and D2 domains of p97, consistent with our crosslinking MS data. However, it should be noted that due to the

disordered nature of most of the VCF1 protein the potential interactions with the p97 D1 and D2 domains were only modeled with low confidence (Figure S3a).

See also our response to Reviewer #3's point 1 below.

3. "2,5 mM Desthiobiotin" should be corrected to "2.5 mM Desthiobiotin." (line 487)

This has now been corrected.

Reviewer #3:

I carefully read the manuscript by Mirsanaye et al. on the regulation of human p97 by cofactors. It is a really interesting manuscript, well written and understandable in every part. P97 is an important player in the proteasome degradation pathway. The authors discover and characterize a novel cofactor, VCF1, which has an unconventional interaction mode and which strongly regulates the activity of p97 in the UFD1-NPL4 pathway. The authors lucidly investigate the interactions between p97 and VCF1, as well the functional modulation, using a battery of methodologies. I fully endorse the publication, and I have a few minor comments and suggestions:

1) The models of VCF1-p97 generated with AF2 are somewhat validated by crosslinking MS, alanine scanning mutagenesis, domain deletion, sequence conservation, and other mutational assays. These models overall are making a lot of sense and are very useful. The authors report that the regions outside the C-terminal part of VCF1 have a very low pLDDT score, and this is in agreement with the intrinsically disordered character of the protein. Of course the lack of confidence does not guarantee for lack of structure, and in fact cross-links are also present between the N-terminal and p97. It could be interesting to see if AF can predict any structure modeling the central peptide of VCF1 (which heavily crosslinks with the rest of p97) and p97, although I suspect that these are transient interactions, as reported by the authors.

Each AlphaFold run outputs several models ranked by overall score. Reassuringly, the highest ranked models for the p97-VCF1 dimer all predict the interaction between the C-terminal VCF1 α -helix and the p97 N domain as shown in our manuscript, whereas the more N-terminal flexible parts of VCF1 with pLDDT scores below 50 are predicted quite differently between the different models, a common outcome for low confidence regions (a superposition of these models is shown in Reviewer Figure 4 below). Interestingly, all these models indicate that the regions of VCF1 N-terminal to its p97-binding α -helix make contacts with the p97 D1 and D2 domains, in line with our crosslinking MS data, but the precise interaction regions vary between the different models (Reviewer Figure 4), thus no definite conclusions about the nature of contacts involving p97 and VCF1 regions outside its C-terminus can be drawn at this point. As also mentioned by the reviewer, these interactions are expected to be transient given that mutation of the VRM in VCF1 drastically reduces its affinity for p97 (Figure 2f; Figure S2d,e).

2) The AlphaFold results and methods are a bit slim, and it is important to report a bit more details: it would be interesting to see the Sequence coverage, the plot of the pLDDT, and especially the PAE plot for the VCF1-p97 and the full complex, to have an idea of the quality of the assembly performed by AF multimer. How many models are generated? Is

there any remarkable difference between them? Also, all models and results, as well as the input files, from AF should be made available, for instance by depositing in Zenodo.

A pLDDT plot for the VCF1-p97 complex is shown in Figure S3a. We now also include PAE plots for the VCF1-p97 and p97-UFD1-NPL4 complexes (new Figure S3b; new Figure S4i – please note that the model of the VCF1(VRM)-p97-UFD1-NPL4 complex (Figure S4h) was generated by superimposing three copies of the AlphaFold-Multimer prediction of monomeric VCF1(VRM)-p97 onto three p97 subunits within the AlphaFold-Multimer model of the hexameric p97-UFD1-NPL4 complex, as described in the Results section (page 11) and the legend for Figure S4h). We generated up to 5 models of the VCF1-p97 complex; as described in the preceding point, these models all agree on the conformation of the interaction between the VCF1 VRM and p97 N domain, whereas predicted interactions between VCF1 regions outside its C-terminus and the p97 D1 and D2 domains vary between the different models (Reviewer Figure 4).

We have expanded the description of the AlphaFold modeling analyses in the Methods section (page 31-32 in the revised manuscript). In addition, the AlphaFold models, results and input files have been uploaded to, and are fully accessible at, the Electronic Research Data Archive at University of Copenhagen (https://sid.erda.dk/share_redirect/h6SgLgmOGi). Details on how to access these data are provided in the Data Availability statement in the Methods section (page 35-36 in the revised manuscript).

Reviewer Figure 4. Superposition of the five top-ranked AlphaFold-Multimer models for the VCF1-p97 complex

AlphaFold-Multimer predictions (top five models) of the complex between full-length VCF1 and a p97 monomer, color-coded by pLDDT value (left) or AlphaFold-Multimer model (right).

3) Can the authors report and comment on the intra-molecular crosslinks, ie, those occurring within p97 and within VCF1? What is the accuracy of crosslinks if they are mapped on p97 hexameric structure? They should provide the cross-link database available to the reader.

While our crosslinking mass spectrometry analysis (Figure 3e) identified multiple intramolecular crosslinks in both p97 and VCF1, we did not display these in the original submission for simplicity. In our revised study, we now show these intramolecular crosslinks (marked in light purple) in Figure 3e. The intramolecular crosslinks we identified in p97

agree well overall with previously published crosslinking MS data for p97 (see Figure 1c in ¹¹). The intramolecular crosslinks in VCF1 mainly involve its intrinsically disordered region, consistent with residues in unstructured regions being *a priori* more accessible to crosslinking than residues embedded in folded structures.

The crosslinking MS data have been uploaded to and are freely available at the ProteomeXchange Consortium. Details on how to access these data are provided in the Data Availability statement in the Methods section (page 35-36 in the revised manuscript).

4) In Suppl fig #, paned A, the model confidence scale reads “Very low (pLDDT>50)” instead it should be “Very low (pLDDT<50)”

This error has now been corrected. We thank the reviewer for bringing this to our attention.

References

1. Ye, Y., Tang, W.K., Zhang, T. & Xia, D. A Mighty "Protein Extractor" of the Cell: Structure and Function of the p97/CDC48 ATPase. *Front Mol Biosci* **4**, 39 (2017).
2. Hart, T. et al. High-Resolution CRISPR Screens Reveal Fitness Genes and Genotype-Specific Cancer Liabilities. *Cell* **163**, 1515-26 (2015).
3. Bekker-Jensen, D.B. et al. An Optimized Shotgun Strategy for the Rapid Generation of Comprehensive Human Proteomes. *Cell Syst* **4**, 587-599 e4 (2017).
4. Fujisawa, R., Polo Rivera, C. & Labib, K.P.M. Multiple UBX proteins reduce the ubiquitin threshold of the mammalian p97-UFD1-NPL4 unfoldase. *Elife* **11**(2022).
5. Kroning, A., van den Boom, J., Kracht, M., Kueck, A.F. & Meyer, H. Ubiquitin-directed AAA+ ATPase p97/VCP unfolds stable proteins crosslinked to DNA for proteolysis by SPRTN. *J Biol Chem* **298**, 101976 (2022).
6. Twomey, E.C. et al. Substrate processing by the Cdc48 ATPase complex is initiated by ubiquitin unfolding. *Science* **365**(2019).
7. Deegan, T.D., Mukherjee, P.P., Fujisawa, R., Polo Rivera, C. & Labib, K. CMG helicase disassembly is controlled by replication fork DNA, replisome components and a ubiquitin threshold. *Elife* **9**(2020).
8. Williams, C., Dong, K.C., Arkinson, C. & Martin, A. The Ufd1 cofactor determines the linkage specificity of polyubiquitin chain engagement by the AAA+ ATPase Cdc48. *Mol Cell* **83**, 759-769 e7 (2023).
9. Kondo, H. et al. p47 is a cofactor for p97-mediated membrane fusion. *Nature* **388**, 75-8 (1997).
10. Beuron, F. et al. Conformational changes in the AAA ATPase p97-p47 adaptor complex. *EMBO J* **25**, 1967-76 (2006).
11. van den Boom, J. et al. Targeted substrate loop insertion by VCP/p97 during PP1 complex disassembly. *Nat Struct Mol Biol* **28**, 964-971 (2021).

REVIEWERS' COMMENTS

Reviewer #1 (Remarks to the Author):

The authors have adequately addressed my concerns. During the revision, another study was published in which VCF1/FAM104A was identified as a co-factor of p97 (<https://doi.org/10.7554/eLife.92409>). While the present and published study are partly overlapping (characterization of interaction, nuclear localization), the two studies are largely complementary with the published paper focusing on the nuclear localization and possible role of p97/VCF1 at the chromatin and the current study giving more mechanistic insights in the role of VCF1 in ubiquitin binding and ubiquitin-dependent proteasomal degradation. The authors do not at any point acknowledge this recent study. It is appropriate and in the interest of the readers that the recent study is cited and, where appropriate, discussed in relation to the new findings. Notably, the renaming of FAM104A to VCF1 refers in UniProt database to the earlier publication. I assume that the renaming has been coordinated with the authors of the earlier study (unless the two groups coincidentally came up with the same new name). This should be acknowledged when introducing the new name, for example, by citing the earlier paper.

Reviewer #2 (Remarks to the Author):

This article reports the discovery of a novel p97 binding partner, VCF1, that regulates the ubiquitin degradation by UFD1/NPL4. Despite the unclear mechanism due to the lack of experimental structural data, the authors identify the interaction region through biochemical experiments and propose a plausible mechanism for VCF1-mediated function. The paper is well written and its findings are compelling and interesting. The authors have addressed all of my comments in their revised manuscript and I support publication.

Reviewer #3 (Remarks to the Author):

The authors answered to all the points.